# Voltage sensor conformations induced by LQTS-associated mutations in hERG potassium channels

Aaron N. Chan [1], Co D. Quach [1], Lucas J. Handlin [2], Erin N. Lessie[2], Emad Tajkhorshid [1] & Gucan Dai [2] ✉

Voltage sensors are essential for electromechanical coupling in hERG K[+] channels, critical to cardiac rhythm. These sensors respond to membrane potential changes by moving within the transmembrane electric field. Mutations in hERG voltage-sensing arginines, associated with Long-QT syndrome, alter channel gating, though underlying mechanisms remain unclear. Using live-cell fluorescence lifetime imaging microscopy, transition metal FRET, an improved dual stop-codon-mediated strategy for noncanonical amino-acid incorporation, and molecular dynamics simulations, we identify intermediate voltage-sensor conformations induced by neutralizing key arginines in the charge transfer center. Phasor plot analysis of lifetime data reveals multiple voltage-dependent FRET states in these mutants, in contrast to the single high-FRET state observed in controls. These intermediate FRET states reflect distinct conformations of the voltage sensor, corresponding to predicted structures of voltage sensors in molecular dynamics simulations. This study provides insights into cardiac channelopathies, highlighting a structural mechanism that impairs voltage sensing in cardiac arrhythmias.

Voltage sensing of ion channels is crucial for regulating cardiac and neuronal excitability[1]. Voltage sensors detect changes in membrane voltage, thereby initiating conformational changes that govern the opening and closing of voltage-gated ion channels (VGICs)[1,2]. This pivotal role allows VGICs to precisely control the flux of ions across cell membranes, thereby orchestrating essential physiological processes such as action potentials in neurons, muscle contraction, cardiac pacemaking, and secretion of neurotransmitters. Notably, dysfunction of ion channels is implicated in various disease states, including the prolonged QT-interval syndrome (long QT syndrome, LQTS), a cardiac disorder characterized by prolonged ventricular repolarization, which can cause life-threatening arrhythmias. Mutations in the human *ether-à-go-go*-related gene (hERG), which encodes the cardiac potassium channel Kv11.1, represent a prevalent cause of congenital LQTS[3,4]. The

hERG channel, belonging to the VGIC superfamily, features a functional voltage sensor, where mutations linked to LQTS have been identified[5].

Voltage sensors of VGICs are characterized by repeats of positively charged arginines within the 4th transmembrane helix (S4), which directly sense the transmembrane electric field[2]. The function of the S4 relies on its significant "up and down" movements within the voltage-sensing domain (VSD)[2,6–13]. These movements are central to electromechanical coupling, the process by which electrical changes across the membrane are converted into mechanical motions that open or close the channel pore. In hERG channels, the S4 helix is part of a conserved VSD (S1–S4 helices) and connects to the pore domain (PD; S5–S6 helices) through a short S4-S5 linker[14]. A cryo-EM structure has captured hERG in a conformation with the S4 in the up position and the

[1]Theoretical and Computational Biophysics Group, NIH Center for Macromolecular Modeling and Visualization, Beckman Institute for Advanced Science and Technology, Department of Biochemistry, and Center for Biophysics and Quantitative Biology, University of Illinois at Urbana-Champaign, Champaign, IL, USA. [2]Edward A. Doisy Department of Biochemistry and Molecular Biology, Saint Louis University School of Medicine, St. Louis, MO, USA. ✉e-mail: gucan.dai@health.slu.edu

pore in an open state[14]; however, other conformational states remain largely uncharacterized. While hERG shares a general domain architecture with other VGICs, it also displays distinct features, including a non-domain-swapped VSD−PD arrangement and potential modulation by intracellular regions such as the N-terminal Per-Arnt-Sim (PAS) domain and the C-terminal C-linker/cyclic nucleotide-binding homology domain (CNBHD)[14,15].

The conformational dynamics of the S4 helix−particularly the elusive down state at hyperpolarized voltages−are critical for understanding voltage sensing in VGICs. High-resolution structures of the S4 in its up-state conformation were initially resolved in the absence of voltage difference (0 mV)[2,16,17]. In comparison, capturing the down state, which corresponds to the S4 voltage sensor at hyperpolarized membrane potentials, has been notably more challenging. This down conformation was first observed in an *Arabidopsis* two-pore channel (TPC)[7]. Similar down conformations have been identified in other VGICs such as voltage-gated sodium channels and pacemaker hyperpolarization-activated cyclic nucleotide-gated (HCN) channels[8−11,18]. Furthermore, cryo-electron microscopy (cryo-EM) studies have provided atomic-level insights into voltage sensors under hyperpolarizing conditions. This was achieved by reconstituting the channels into liposome membranes and manipulating ionic gradients to apply hyperpolarizing voltages, an approach that requires the channel pores to remain closed to prevent ion leakage across the membrane[12,13]. This approach has yielded cryo-EM structures of ether-à-go-go (Eag) channels, a close relative of hERG, in multiple conformational states including up, down, and intermediate forms[12]. However, studying the conformational dynamics and heterogeneity of these S4 voltage-sensor states under native membrane environments remains a significant challenge. A more comprehensive understanding of the structural dynamics and potential multi-step movements of the voltage sensor is crucial for elucidating the electromechanical coupling governing VGIC function.

While fluorescence imaging stands as a valuable tool for understanding the structural mechanisms of voltage sensors, significant technical hurdles arise in controlling membrane voltage for such imaging experiments. Single-molecule fluorescence techniques hold promise, with their capacity to discern distinct protein conformational states, yet their application in probing the voltage-dependent structural arrangements of VGICs remains limited in a native membrane[19,20]. Understanding the structure-function relationship of VGICs often necessitates live-cell imaging, as dynamics of voltage sensors are influenced by membrane lipids in their native environments. However, traditional intensity-based ensemble methods average signals across molecules, often obscuring conformational heterogeneity. In contrast, fluorescence lifetime imaging microscopy (FLIM), especially when combined with transition metal Förster resonance energy transfer (tmFRET), can distinguish structural states based on fluorescence lifetime rather than intensity[21−24]. This study introduces an application of FLIM-tmFRET with site-specific incorporation of noncanonical amino acids (ncAAs) to monitor conformational dynamics of the S4 in live cells. Our approach enables high-resolution, state-specific measurements under near-native conditions, offering a significant advance in the structural study of voltage sensors within VGICs.

In this study, we employ a combination of techniques including dual stop-codon suppression-mediated incorporation of ncAAs for site-specific fluorescence labeling, phasor FLIM-tmFRET, and molecular dynamics simulations to examine the impact of a LQTS-associated mutation on conformational rearrangements of hERG voltage sensors in transfected human cell lines. We identify structural alterations in hERG voltage sensors, including low-energy conformational intermediates, that underlie the functional effects of LQTS-associated mutations. This integrative approach offers a powerful and broadly applicable platform for quantitatively assessing the movements, conformational states, dynamics, and energetics of voltage sensors in ion channels.

## Results

### Functional impact of S4 helix LQTS mutations on hERG channel activation

To investigate how specific charged residues in the S4 helix contribute to voltage sensing and gating in hERG channels, we focused on a subset of mutations previously implicated in LQTS. The hERG ($K_v$11.1 or KCNH2) channels expressed in tsA-201 cells exhibit a depolarization-dependent activation profile like other voltage-gated potassium channels[1]. Within the S4 helix of hERG, six positively charged arginines or lysines are present, with the arginines within the charge transfer center presumably serving as the primary voltage-sensing residues[25] (Supplementary Fig. 1a). Notably, most of these charged residues have been implicated in LQTS[5,26,27] (Supplementary Fig. 1a).

Homology models of hERG were generated using SWISS-MODEL, guided by previous cryo-EM studies that identified three distinct conformations of the voltage sensor in the rat Eag channel[12,28]. In the down-state model at hyperpolarized voltages (Supplementary Fig. 1b), only the side chain of K525 of hERG is positioned above the "phenylalanine cap", F463, at the hydrophobic constriction site[25]. Upon membrane depolarization, the side chains of K525, R528, and R531 are predicted to relocate above the phenylalanine cap. In comparison, in the model of the hypothetical intermediate state, the side chains of K525 and R528, but not that of the R531, are positioned above the phenylalanine cap. This intermediate conformation may represent a transitional step during voltage sensor movement in hERG channels.

We examined the functional consequences of charge-neutralizing mutations in the context of the non-inactivating S620T background. In the presence of 10 mM tetraethylammonium (TEA) to fully inhibit endogenous potassium currents in tsA cells[29] (Supplementary Fig. 1c)− different from the wild-type (WT+; with S620T) hERG channels[30] - hERG channels with LQTS-associated mutation R531Q (R531Q+; also with S620T) demonstrated a noticeable initial phase of approximately 100 ms, preceding the delayed second phase (Fig. 1a). The activation profile was plotted using the instantaneous "tail current" amplitudes at −100 mV during repolarization at the end of the voltage protocol (Fig. 1a, b). The S620T mutation, near the selectivity filter, abolishes C-type inactivation in hERG channels, thereby enabling analysis of channel activation independently of the inactivation[30−32]. In this context, tail current amplitudes recorded from R531Q+ mutant channels revealed a substantial rightward shift in the conductance−voltage (G−V) relationship compared to wild-type channels (Fig. 1b); the voltage for the half maximal activation of R531Q+ channels was +81 ± 3.7 mV ($n = 4$) compared to +35 ± 2.8 mV ($n = 8$) for the WT+ channel. This large ~45 mV shift in the G-V and the biphasic channel activation are consistent with prior research and indicative of potential intermediate states of the S4[26,33,34]. Furthermore, this initial lag was less noticeable in hERG-R528Q+ (with S620T) channels, followed by a double-exponential main current phase for the activation; the deactivation was much slower (Supplementary Fig. 1d)[35]. The R528Q mutation did not significantly shift the G−V relationship ($V_{1/2} = +24 ± 7.7$ mV, $n = 4$) compared to the WT+ and has not been linked to LQTS. Additionally, another LQTS-linked mutation of the S4 arginines R534L[26] (with R534Q also tested) produced hERG channels that have diminished multi-phasic activation kinetics. Instead, these channels exhibited considerable voltage-induced channel inactivation, despite the elimination of C-type inactivation by the S620T mutation (Supplementary Fig. 1e). Collectively, assessing the functional impact of these three charge-neutralization mutations on channel activation, the R531Q produces the most significant effect. This result is consistent with the finding that R531 is the primary residue that contributes to the total gating charge of 1.5 elementary electronic charges per subunit−a value notably smaller than that of classical voltage-gated potassium channels[33,36,37].

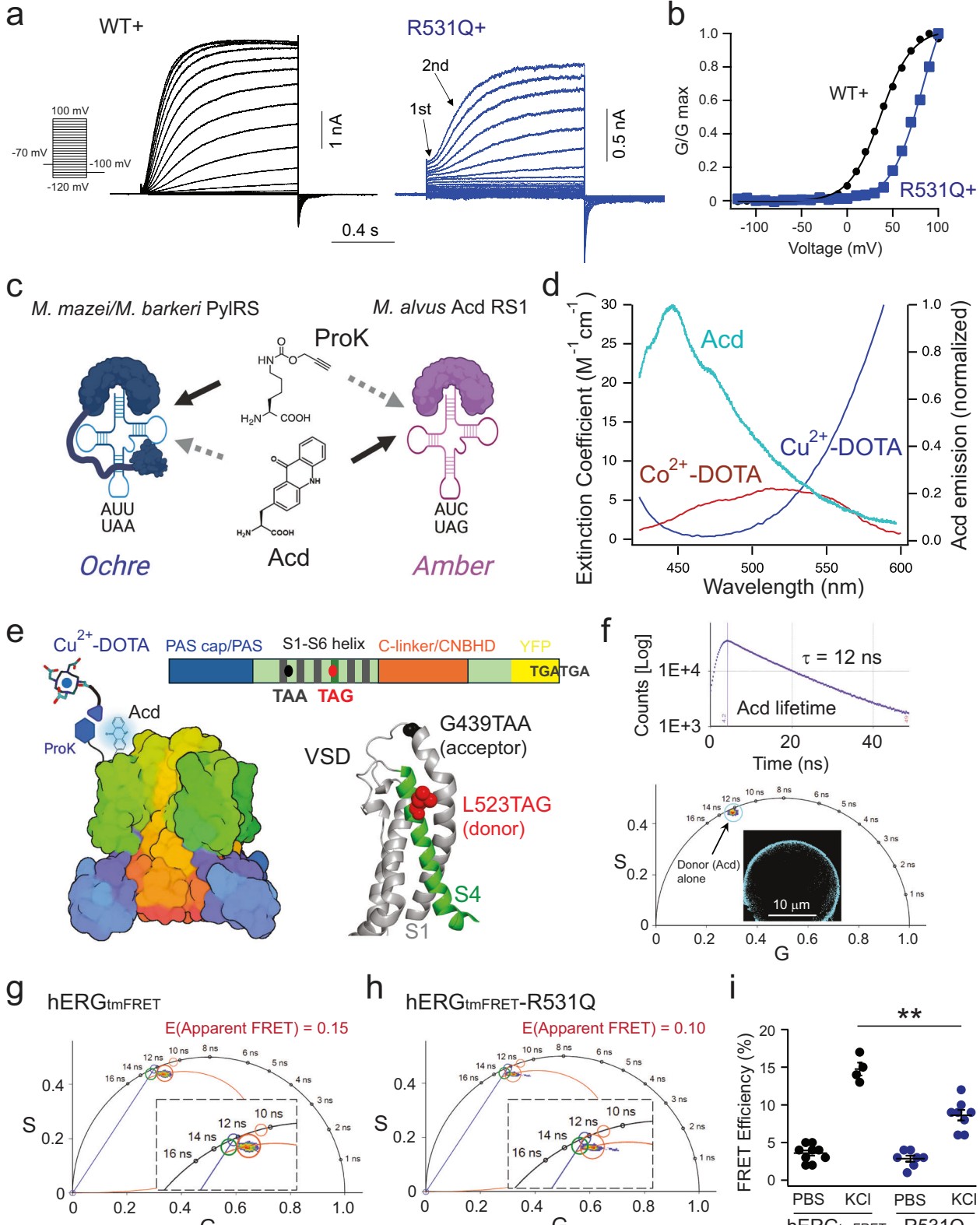

## Measuring the movement of voltage sensors in hERG channels using FLIM-tmFRET

To directly visualize conformational changes in the VSD of hERG channels and assess how LQTS mutations affect these dynamics, we implemented a dual stop-codon suppression strategy to incorporate noncanonical amino acids for site-specific fluorescence labeling and FRET measurements of voltage-sensor movements[38]. We modified a

human hERG1 construct by introducing amber (TAG) and ochre (TAA) stop codons at selected positions within the VSD of hERG channels (Fig. 1). The amber stop-codon suppression approach facilitated the incorporation of a fluorescent noncanonical amino acid L-acridonyl-alanine (Acd) into the S4 helix[39,40]. This was achieved using a recently developed Acd tRNA/tRNA synthetase system adapted from the pyrrolysyl-tRNA synthetase of the

**Fig. 1 | Design of phasor FLIM-tmFRET strategies for measuring voltage sensor movement in hERG channels. a** Representative current traces for activation of WT + and R531Q+ hERG1 channels, elicited by steps of depolarizing voltage pulses from −120 to +100 mV. Tail currents were recorded at −100 mV. **b** Comparison of hERG channel activation G–V curves, derived from recordings in **a** WT+ (circles) vs. R531Q + (squares). **c** Diagram illustrating the dual stop-codon (amber/ochre) suppression for incorporating Acd and ProK in a mutually orthogonal manner. Cartoons were created in BioRender. Dai, G. (2025) https://BioRender.com/1x7gojd. **d** Absorption and emission spectra showing spectral overlaps between the Acd emission incorporated to hERG channels (L523 site) and the molar absorption of $Co^{2+}$ or $Cu^{2+}$ coordinated with azido-DOTA. **e** Design of a tmFRET pair with the FRET donor positioning in the L523TAG site of the S4 helix and the FRET acceptor in the C-terminal end of the S1 helix (near the S1-S2 linker). YFP was not shown in the structural cartoon. Created in BioRender. Dai, G. (2025) https://BioRender.com/

zs1deap. **f** Mono-exponential lifetime decay of Acd when incorporated into the L523 site. The lifetime species of membrane localized Acd can also be displayed in the phasor plot, with $\tau(\varphi) = 11.7$ ns and $\tau(m) = 12.6$ ns. **g** FRET trajectory calculation enabled by the phasor plotting for hERG$_{tmFRET}$ channels, labeled with $Cu^{2+}$ acceptor via CuAAC click chemistry in the presence of 120 mM extracellular KCl. The inset shows an amplified view. **h** Similar FRET trajectory calculation for hERG$_{tmFRET}$-R531Q channels as a comparison. The red cursor covering the lifetime species is more distant from the green cursor of donor alone, indicating greater FRET in this case. **i** Summary data showing the apparent FRET efficiency of hERG$_{tmFRET}$ in regular PBS buffer ($n = 10$ cells) or with 120 mM KCl ($n = 4$ cells), and hERG$_{tmFRET}$-R531Q channels in PBS ($n = 7$) or high KCl ($n = 8$), using the FRET trajectory method as in (**g**, **h**). Data shown are mean ± s.e.m., **$p$ = 2e-6 under high KCl conditions, one-way ANOVA (no adjustment). Source data are provided as a Source Data file.

archaeon *Methanomethylophilus alvus (M. alvus)*[40] (Fig. 1c). Importantly, this orthogonal strategy operates independently of the ochre stop-codon suppression system, which is derived from the pyrrolysyl-tRNA synthetase of the *Methanosarcina* archaea *M. mazei/M. barkeri*[41]. Specifically, we used a tRNA (M15-PylT)/PylRS pair derived from *M. mazei/M. barkeri* to introduce N-propargyl-L-lysine (ProK) into another relatively immobile site within the VSD. ProK features an alkyne group capable of undergoing specific chemical reactions with azide-containing compounds via copper(I)-catalyzed azide-alkyne cycloaddition (CuAAC) click chemistry. Previously, we have applied a similar approach to perform FLIM-tmFRET experiments for HCN channels[38,41], with the integration of Acd marking a notable enhancement over our prior dual stop-codon suppression methodology. Furthermore, this protein labeling design circumvents the necessity for cysteine-based labeling, mitigating potential concerns of mutating endogenous cysteine residues ( ˜160) within hERG channels.

We designed a tmFRET pair comprising Acd as the FRET donor and $Cu^{2+}$ chelated by 1,4,7,10-tetraazacyclododecane-1,4,7,10-tetraacetic acid (DOTA) as the FRET acceptor. The emission spectrum of Acd overlaps with the absorption spectrum of $Cu^{2+}$ or $Co^{2+}$ coordinated by an azido mono-amide-DOTA (Fig. 1d). The weak molar absorption of transition metals results in a shorter Förster distance ($R_0 = $ ˜10−20 Å, depending on the exact design) for tmFRET compared to conventional FRET pairs[23,24,42–44]. This reduced working distance of tmFRET is well-suited for investigating voltage-sensor movement in ion channels[10,11,38,45]. For probing the S4 movement of hERG channels, we positioned the Acd site at the L523 position near the N-terminal side of the S4 helix and the ProK/$Cu^{2+}$-DOTA site at the G439 position at the C-terminal end of the S1 helix (Fig. 1e). This design is predicated on the premise that the FRET donor and acceptor should be distant enough when the S4 is in the down state, resulting in minimal FRET, and exhibit increased FRET when they approach due to membrane depolarization. Because of the short $R_0$ of this tmFRET pair, FRET occurs primarily within the same subunit; inter-subunit FRET is negligible[10,11,38]. Additionally, this hERG construct—designated hERG$_{tmFRET}$ and containing the mutations G439TAA, L523TAG, and S620T—includes a C-terminal YFP tag to indicate the expression of full-length proteins at the cell membrane. Our observations revealed a correlation between YFP fluorescence and Acd fluorescence, confirming the expression of hERG channels (Supplementary Fig. 2). Functionally, for hERG channels with Acd and ProK incorporated (hERG$_{tmFRET}$ construct), similar depolarization-dependent activation was observed, with R531Q significantly rightward shifted the G-V relationship compared to the control (Supplementary Fig. 2).

Phasor FLIM was used to quantify the tmFRET by measuring the lifetime of Acd in tsA cells. One advantage of Acd is its exceptionally long and mono-exponential lifetime ( ˜16 ns free in solution and

˜12 ns when incorporated in cells)[23]. Both time-domain and frequency-domain types of FLIM were performed, which yielded similar lifetime values for the membrane-localized fluorescence coming from the Acd incorporated at the L523 position of the S4 helix (Fig. 1f). In frequency-domain FLIM analysis, lifetime data from each pixel of the confocal image are mapped onto a phasor plot—a complex plane where the universal semicircle serves as a reference for fluorescence lifetimes (Fig. 1f)[21,46,47]. Following Fourier transformation, the x-axis represents the real (G) component, and the y-axis represents the imaginary (S) component of the fluorescence decay. Pixels corresponding to mono-exponential lifetimes fall precisely on the semicircle, whereas multi-exponential decays localize within it. Importantly, shorter lifetimes are positioned further clockwise along the semicircle. Accounting for minor background contributions, the measured Acd phasor in cells localizes close to the semicircle, indicating a near mono-exponential decay, which facilitates our subsequent FRET analysis.

We used the FRET trajectory function facilitated by the phasor approach of FLIM to assess the change in tmFRET, comparing tsA cells expressing hERG channels imaged in either regular phosphate-buffered saline (PBS) or a depolarizing solution containing 120 mM KCl (Fig. 1g–i). This function relies on the concept that the degree of shortened donor lifetime reports FRET efficiency, thereby representing the distance between the FRET donor and acceptor. The FRET trajectory starts from zero FRET efficiency, aligning with the donor-alone reference, and follows a path through the measured phasor species, ending at the theoretical 100% FRET efficiency at the origin (0, 0) coordinate, where most of the background noise is situated (Supplementary Fig. 2b and Supplementary Fig. 3a). For hERG$_{tmFRET}$ channels, FRET efficiency was minimal in PBS as expected but increased upon exposure to 120 mM KCl. In addition, treatment of 120 mM KCl did not affect the lifetime of the Acd donor alone. Conversely, in comparison to the hERG$_{tmFRET}$, the presence of R531Q produced a similarly low FRET efficiency in PBS, but the increase in FRET after the high KCl was attenuated (Fig. 1i). This outcome corresponds to the rightward shift observed in the G-V curve of the hERG-R531Q+ channels, indicating a compromise in the extent or likelihood of voltage sensor movement.

However, while the FRET trajectory method used in ensemble imaging experiments offers valuable insights, it also comes with limitations. Although it provides a quantified FRET efficiency, it only represents an averaged distance between donors and acceptors, lacking the capacity to reveal more detailed structural information and the conformational heterogeneity of the S4 helix (Supplementary Fig. 3a). As a result, this approach is insufficient for testing hypotheses about intermediate voltage sensor states or assessing how specific arginine mutations may influence distinct conformational states. To address these limitations, we employed an alternative strategy based on the principles of phasor addition and linear unmixing in phasor FLIM[21,47].

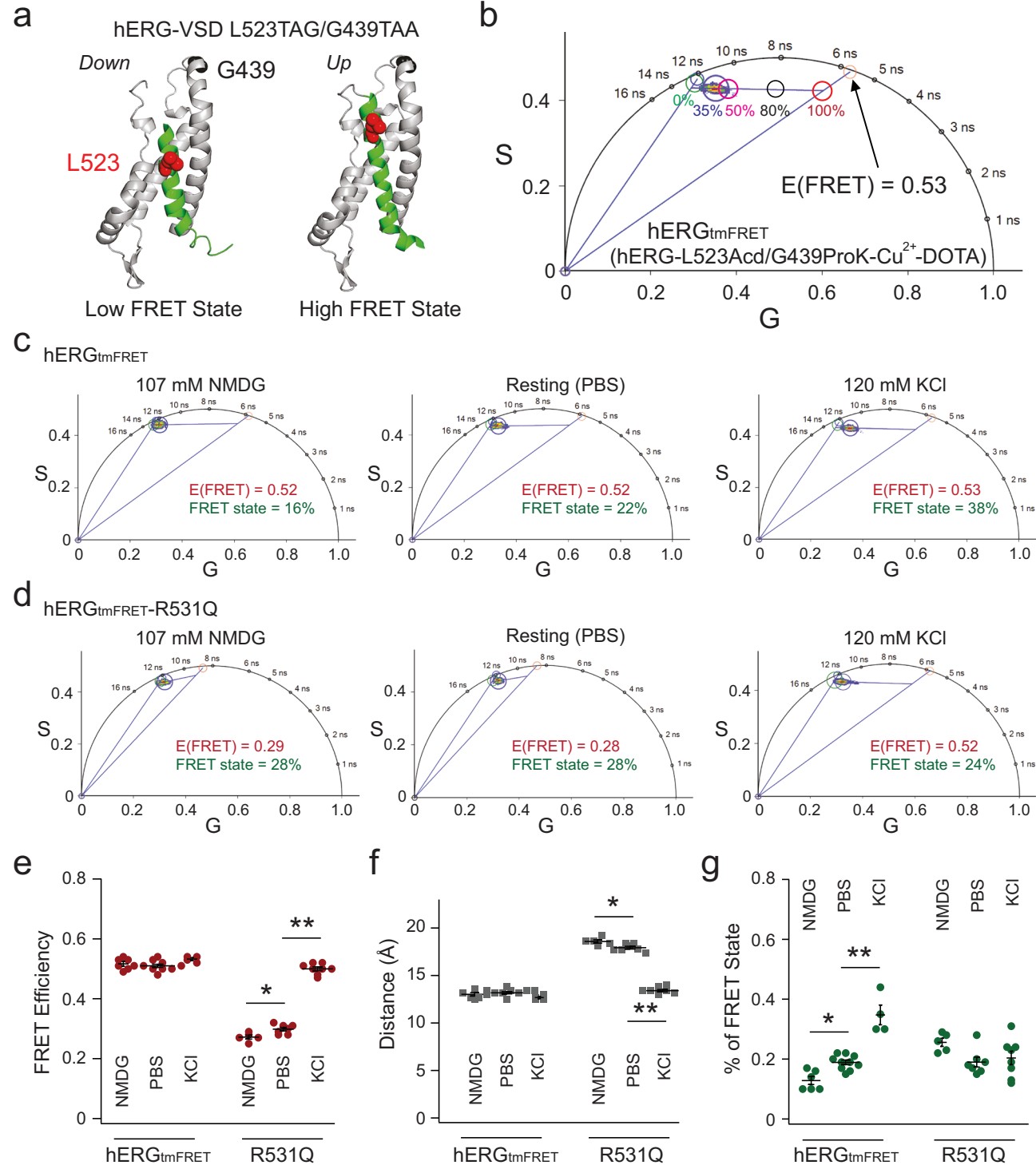

## Probing the conformational heterogeneity of hERG voltage sensors using the linear unmixing approach of phasor FLIM

This alternative approach takes advantage of the linear unmixing capability of the FLIM phasor plot to resolve and quantify the fractional populations of distinct FRET states[21,22,46,47]. We consider the voltage sensor as existing in two conformational states: up and down states. The down-state model of the S4 corresponds to the low FRET state, while the up-state model corresponds to the high FRET state (Fig. 2a). In our tmFRET design, the low FRET state—arising when the donor–acceptor distance exceeds ~25 Å—results in minimal energy transfer, leaving the donor lifetime nearly identical to that of the donor alone. As a result, the overall FRET signal is primarily determined by

the efficiency of the high FRET state and the probability of the voltage sensor occupying that state.

To demonstrate linear unmixing in the phasor plot, we used cells expressing the hERG-L523TAG/G439TAA construct, dually labeled with Acd and Cu²⁺-DOTA (Fig. 2b). Under depolarizing conditions achieved with 120 mM external KCl, the measured FRET can be resolved into two contributors: the donor only and the position of lifetime if 100% of the voltage sensors are in the high FRET state (Fig. 2b and Supplementary Fig. 3b). In a phasor plot, lines that connect different lifetime species are linear combination lines. The position of a phasor point along the line reflects the fractional contribution of each species in the mixture[46]. Here, linear fitting of the phasor data reveals a

**Fig. 2 | Applying linear unmixing of phasor analysis to the FLIM-tmFRET for studying hERG channels. a** Structural cartoons illustrating the hypothetical low and high FRET states in the context of L523TAG/G439TAA FRET pair, corresponding to homology models of hERG-VSD in the down and up states. Distances between the β carbon of L523 and the α carbon of G439, D(Cβ-Cα), are 29 Å and 19 Å in the down and up states, based on homology hERG models derived from cryo-EM structures of Eag channels. **b** Representative linear unmixing approach to estimate the FRET efficiency corresponding to the high FRET state of hERG$_{tmFRET}$ channels at 120 mM KCl. The red cursor points to a hypothetical scenario that 100% of voltage sensors are in the up position, and thus the high FRET state of 35%. The arrow points to the cursor of the lifetime associated with the high FRET-state if background contribution is absent. Representative phasor-based tmFRET analysis from individual cells expressing hERG$_{tmFRET}$ (**c**) and hERG$_{tmFRET}$-R531Q (**d**) channels, using the L523/G439 pair. FRET was measured under three conditions: NMDG (to induce

membrane hyperpolarization), dPBS (reference condition), and 120 mM KCl (to induce membrane depolarization). In hERG$_{tmFRET}$ (**c**), FRET efficiency is similar across all conditions, whereas in hERG$_{tmFRET}$-R531Q (**d**), distinct FRET efficiencies are observed. Summary data showing FRET efficiencies (**e**), estimated donor–acceptor distances (**f**), and the fraction of the high FRET state (**g**), calculated via linear unmixing using the phasor approach. For hERG$_{tmFRET}$-R531Q, statistical comparisons of FRET efficiency and distance yielded $p = 0.03$ (NMDG vs dPBS), $p = 3.5e-13$ (NMDG vs KCl), and $p = 3.8e-13$ (dPBS vs KCl), by one-way ANOVA (no adjustment). For hERG$_{tmFRET}$, statistical comparisons of the percentage of FRET yielded $p = 0.2$ (NMDG vs dPBS), $p = 1.4e-7$ (NMDG vs KCl) and $p = 3e-6$ (dPBS vs KCl). Sample sizes: hERG$_{tmFRET}$: NMDG ($n = 6$ cells), dPBS ($n = 10$ cells), KCl ($n = 4$ cells); hERG$_{tmFRET}$-R531Q: NMDG ($n = 5$ cells), dPBS ($n = 7$ cells), KCl ($n = 8$ cells). Data are presented as mean ± s.e.m. Source data are provided as a Source Data file.

trend that follows the major axis of the elliptical cluster formed by membrane-localized pixels in the phasor plot. This axis likely reflects conformational heterogeneity among the channel population, with individual molecules adopting a range of S4 states. A more elongated major axis (i.e., a higher length-to-width ratio) may indicate greater variability in donor–acceptor distances across the channel ensemble. The measured high FRET value of 0.53, corresponding to a converted distance of approximately 13 Å, reflects the FRET efficiency of the high FRET state, and an estimation of the donor–acceptor distance in the up-state model. Based on the law of phasor additions[46], the fractional contributions of different lifetime species are inversely related to their distance from the composite phasor point in the plot—providing a quantitative basis for estimating the proportion of each state present. Therefore, the proportion of S4 segments in the high FRET state can be estimated by calculating the relative distances from the measured phasor position to the donor-only and high FRET reference points along the linear fit. As expected, membrane depolarization increased the proportion of the high-FRET state; approximately 35% of voltage sensors transitioned to the up state, consistent with the relatively low open probability of wild-type hERG channels at around +10 mV—a potential readily achieved by applying 120 mM extracellular KCl, as previously shown using voltage-sensitive dyes[38] (Fig. 2b). To establish a hyperpolarized baseline, we used extracellular NMDG to reduce resting membrane potential[48]. Further depolarization led to a greater fraction of the high-FRET state, with minimal changes in the amplitude of FRET efficiency (Fig. 2c). These results suggest that hERG$_{tmFRET}$ channels primarily occupy two discrete S4 conformations, with limited sampling of intermediate states.

To demonstrate the sensitivity of our phasor FLIM-tmFRET strategy in detecting small distance changes, we used a single mutation, D540K, which induces a "U-shaped" G-V relationship[49]. Notably, hERG-D540K channels exhibit pronounced hyperpolarization-activated currents reminiscent of pacemaker HCN channels, alongside depolarization-activated currents (Supplementary Fig. 4a–d). The D540K mutation disrupts a critical salt bridge between D540 at the C-terminal end of the S4 and R665 in the C-terminal end of the S6 helix (Supplementary Fig. 4a). This disruption leads to the decoupling of the S4 and S6, resembling the structural arrangement observed in HCN channels, where the S4 presumably relies on coupling with the S5 helix rather than with the S6 helix for hyperpolarization-dependent activation[10,11,50]. To restore the wild-type behavior of the channel, we engineered a charge-swapping mutant, hERG-D540K/R665E (Supplementary Fig. 4c,d), to recreate the salt bridge and thereby reestablish the S4-S6 coupling[49]. This intervention successfully rescued the native function of hERG channels.

The distinctive U-shaped G-V profile of the D540K mutation suggests the presence of multiple functional and conformational states of the S4 voltage sensor. To test whether these states correspond to distinct FRET efficiencies, we introduced the D540K mutation into hERG$_{tmFRET}$ channels. In the presence of D540K, phasor analysis

revealed clear differences in tmFRET efficiency under hyperpolarized versus depolarized conditions (Supplementary Fig. 4e, f, Supplementary Fig. 5a, c). In contrast, channels containing double D540K/R665E mutations exhibited only a single high-FRET state (Supplementary Fig. 5b, c). These results suggest that D540K introduces distinct S4 conformations that likely contribute to the U-shaped gating behavior, while the R665E addition restores the all-or-none S4 transitions characteristic of the hERG$_{tmFRET}$ channel (Supplementary Fig. 5). This comparative analysis underscores the sensitivity of FLIM phasor analysis in detecting subtle distance changes between FRET pairs arising from voltage sensor rearrangements.

We also tested a tmFRET pair using the hERG L523TAG/A430TAA construct (Supplementary Fig. 6). In this case, even in the down state of S4, the proximity of the donor and acceptor resulted in noticeable FRET signals. As a result, the Acd lifetime distribution displayed a distinct arc shape, contrasting with the elliptical pattern observed in the hERG L523TAG/G439TAA construct. The primary lifetime distribution after a linear fitting, yielded a notably higher FRET efficiency (~60% efficiency), whereas the minor component of the lifetime distribution exhibited a lower FRET efficiency (Supplementary Fig. 6b–d). The presence of this minor component complicated the accurate fitting of the lifetime distribution, especially when assessing the existence of additional intermediate FRET efficiencies, unless a second harmonic of the light modulation frequency was included[51,52]. Nevertheless, these results emphasize the ability of our FLIM-tmFRET strategy to reliably report S4 movement, with FLIM-tmFRET measurements showing dependence on both donor/acceptor proximity and voltage changes (Supplementary Fig. 6e, f). Because linear unmixing analysis is most effective in systems that approximate a two-state model—especially when one state exhibits minimal FRET, as observed in the L523/G439 FRET pair—we focused on the hERG L523/G439 pair (hERG$_{tmFRET}$ construct) for testing the LQTS mutations.

### Phasor FLIM-tmFRET reveals intermediate FRET states in hERG voltage sensors induced by R531Q

Using the same L523/G439 pair in hERG$_{tmFRET}$, we tested the impact of the LQTS mutation R531Q on voltage sensor conformations (Fig. 2d). Our phasor analysis revealed at least two different FRET efficiencies: a lower one around 0.25 and a higher one similar to those observed in tmFRET experiments using the hERG$_{tmFRET}$ (Fig. 2d, e). These findings suggest the presence of an intermediate conformation of the S4 voltage sensor induced by R531Q, exhibiting increased donor–acceptor distances (Fig. 2f). There was a notable prevalence of voltage sensors in the FRET state, even under more hyperpolarized voltages, with no increase upon voltage elevation in our experimental conditions (Fig. 2g). Given the pronounced right shift in the G-V curve of hERG-R531Q+ channels, a more depolarized voltage (> + 60 mV) might be necessary to observe an increase in the percentage of the FRET state. At hyperpolarized voltages, the voltage sensor primarily transitions between the down and intermediate states relatively frequently

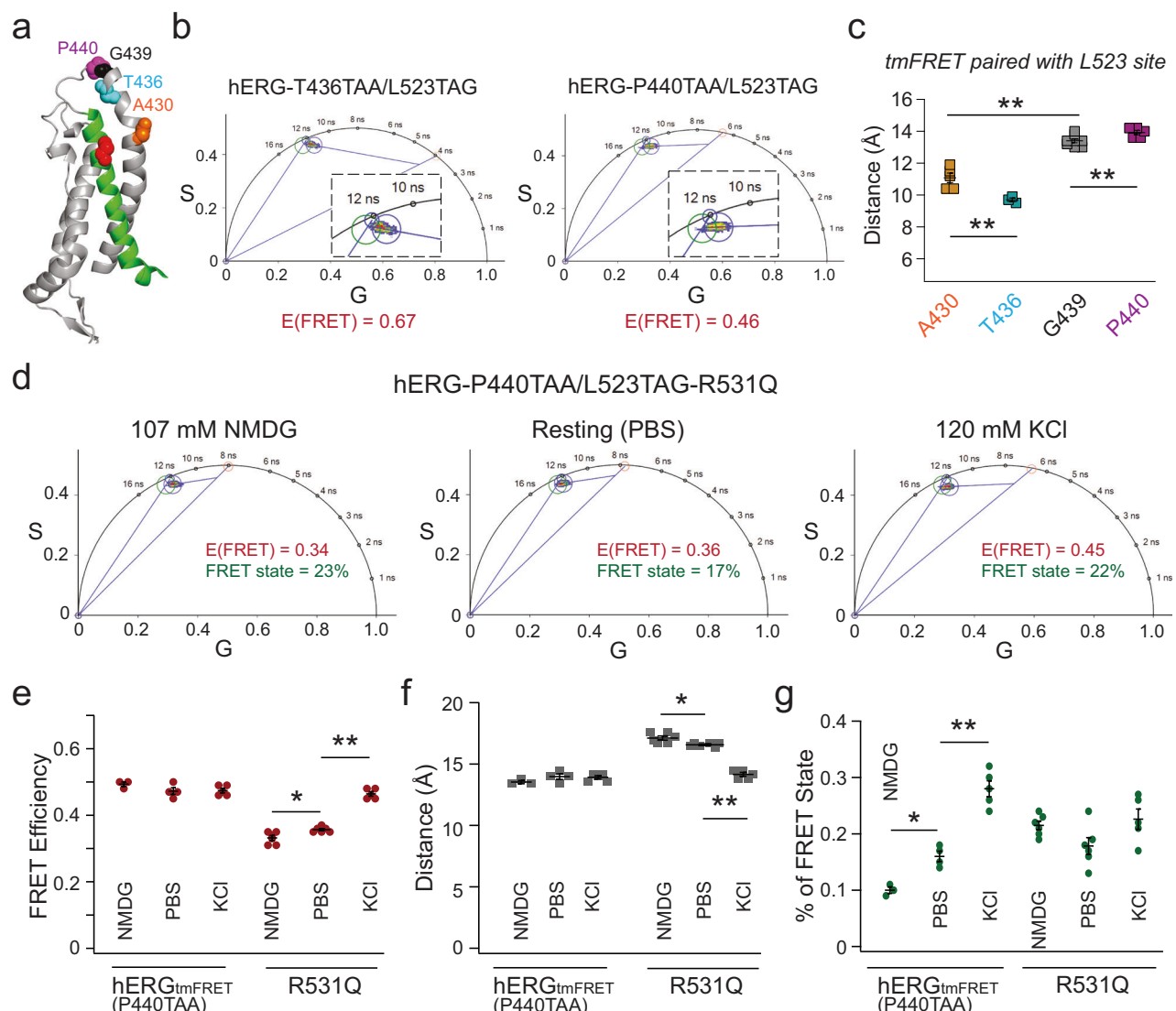

**Fig. 3 | Alternative tmFRET pairs yielded similar intermediate distances in the hERG-R531Q channel. a** Structural cartoon illustrating the four different tmFRET pairs, L523TAG as the donor and A430TAA, T436TAA, G439TAA, P440TAA as acceptors. **b** Representative phasor analysis showing the different FRET efficiencies comparing the L523TAG/T436TAA and L523TAG/P440TAA pairs. Cells are imaged in the 120 mM KCl depolarizing condition. **c** Summary of donor–acceptor distance estimates calculated from measured tmFRET of the four pairs in the depolarizing (high KCl) condition, $n = 5$ cells for A430, $n = 3$ cells for T436, $n = 4$ cells for G439, and $n = 5$ cells for A440. Data are presented as mean ± s.e.m. One-way ANOVA (no adjustment) showed statistical significance between all pair-wise comparisons, particularly $p = 4e\text{-}3$ (A430 vs T436), $p = 5e\text{-}4$ (A430 vs G439), $p = 4e\text{-}3$ (G439 vs P440). **d** Representative phasor-plot analysis of the L523/P440 tmFRET from

individual cells for hERG$_{tmFRET}$ (control) and hERG$_{tmFRET}$-R531Q channels under various conditions. Summary data showing FRET efficiencies of hERG$_{tmFRET}$-L523/P440 pair (**e**), estimated donor–acceptor distances (**f**), and the fraction of the high FRET state (**g**). Data shown are presented as mean ± s.e.m. For FRET efficiencies and estimated distances of R531Q, $p = 0.025$ (NMDG vs dPBS), $p = 1.3e\text{-}9$ (NMDG vs KCl), and $p = 2.1e\text{-}8$ (dPBS vs KCl), one-way ANOVA (no adjustment). For the percentage of FRET state of hERG$_{tmFRET}$-L523/P440, $p = 0.024$ (NMDG vs dPBS), $p = 7.8e\text{-}6$ (NMDG vs KCl), and $p = 1e\text{-}4$ (dPBS vs KCl), one-way ANOVA (no adjustment). Sample sizes: hERG$_{tmFRET}$: NMDG ($n = 3$ cells), dPBS ($n = 4$ cells), KCl ($n = 5$ cells); hERG$_{tmFRET}$-R531Q: NMDG ($n = 6$ cells), dPBS ($n = 6$ cells), KCl ($n = 5$). Source data are provided as a Source Data file.

(Fig. 2g, and percentage of FRET state = ~20%), while at more depolarized voltages, it can ascend to the up state. The donor–acceptor distance of 19 Å, derived from the measured intermediate FRET efficiency (in comparison to 13 Å for the high FRET state), aligns with the intermediate conformation of VSD predicted by the cryo-EM structure of Eag channels[12].

To further validate the robustness of our method and test the hypothesis that R531Q produces additional intermediate conformational states of S4, we designed alternative FRET pairs. Using the same FRET donor site, L523TAG, we selected two additional acceptor sites: T436, and P440 (Fig. 3a). Both constructs, paired with L523TAG, exhibited noticeable membrane-localized YFP expression after employing the dual stop-codon suppression strategy. Compared to

the L523/G439 FRET pair, the L523/P440 pair demonstrated lower FLIM-tmFRET efficiency, whereas the L523/T436 pair exhibited higher efficiencies in the presence of 120 mM KCl (Fig. 3b). Overall, the converted distances for the four FRET pairs exhibited differences consistent with their respective positions in our structural models, despite potential variability introduced by donor–acceptor rotamer conformations inherent to this type of FRET measurement (Fig. 3c and Supplementary Fig. 7a). In addition, linear regression analysis revealed a strong negative correlation (coefficient $r = -0.88$) between the measured FRET efficiencies and the backbone-atom distances of these FRET pairs in hERG homology models with S4 in the up state. Regarding the L523/P440 pair, similar to the FRET changes observed with the L523/G439 pair, only a single high-FRET state was detected,

with the percentage of this state adjustable by altering membrane voltage using external NMDG or 120 mM KCl (demonstrated with the L523/P440 pair in Fig. 3e–g). Moreover, the results obtained with the L523/P440 FRET pair in the presence of R531Q further support our conclusion that the R531Q mutation induces intermediate conformational states of the S4 helix in hERG channels (Fig. 3d). The observation of intermediate FRET efficiencies under less depolarized membrane conditions (such as NMDG and PBS) aligns with the findings from the L523/G439 pair and reinforces the notion that R531Q leads to distinct voltage sensor conformations (Fig. 3e–g). Additionally, since the L523/T436 pair shared the same limitation as the previously mentioned L523/A430 pair, we opted not to test the effect of R531Q on potential intermediate FRET efficiencies for this pair.

One major advantage of our phasor linear unmixing FLIM-tmFRET method over traditional ensemble FRET is its ability to quantitatively estimate the fractional population of distinct FRET or conformational states[24,53]. Using this approach, we determined that approximately 35% of the S4 in hERG$_{tmFRET}$ channels adopt the high FRET state under high KCl conditions, compared to only 20% in the hERG$_{tmFRET}$-R531Q channels. We found a tetrameric model where at least three subunits must be activated for pore opening fitted our data best, compared to models requiring either one, two, or four subunits to be activated (see Methods and Supplementary Table 1). The corresponding Boltzmann free energy changes (ΔG) for tetramer activation of hERG were 1.1 kcal/mol for WT and 2.4 kcal/mol for R531Q. Notably, these estimates are close to ΔG values independently derived from patch-clamp G-V relationships, which yielded 1.3 kcal/mol (WT) and 2.9 kcal/mol (R531Q). This consistency highlights the effectiveness of our FLIM-tmFRET approach in extracting energetically meaningful parameters of conformational equilibria.

Collectively, these results indicate that: (1) the FLIM signals we measured predominantly originate from membrane-localized hERG channels; (2) dual stop-codon suppression enables the incorporation of Acd and ProK, yielding functional VSDs that respond to membrane potential changes; and (3) The phasor linear unmixing FLIM-tmFRET method provides accurate measurements of short-range distance changes between membrane helices in a near-native lipid environment, while simultaneously enabling quantification of conformational state populations and associated free-energy differences.

## Molecular dynamics simulations suggest R531Q induces distinct conformations of the S4

We performed molecular dynamics (MD) simulations to gain structural insights into the conformational dynamics and heterogeneity of the hERG VSD. When a hERG channel tetramer, modeled based on the cryo-EM structure of the Eag channel in the down conformation, was relaxed in a native lipid environment at −150 mV using MD simulations[54], we observed that the S4 helix adopted the down conformation in two out of the four VSDs. In the other two VSDs, however, it adopted an intermediate conformation, similar to the intermediate-state cryo-EM structure of Eag-channel VSD and the intermediate I state inferred from tmFRET results (Supplementary Fig. 8), providing initial MD-based evidence of intermediate state. When we repeated the simulations at −150 mV and 0 mV for a monomeric hERG transmembrane domain, the VSD was also stabilized in the intermediate I conformation (Fig. 4a). Previous studies have shown that the VSD of KCNH channels can function as a relatively independent unit[55,56]. This is exemplified in hERG and Eag channels, where split constructs with covalently disconnected VSDs and pore domains can reassemble into functional channels[55]. For Eag channels, even when both the entire N- and C-terminal regions are deleted, the channels still retain normal depolarization-dependent activation properties[56].

To evaluate how the R531Q mutation alters the resting conformation of the hERG VSD, we simulated the relaxed state of the mutant hERG VSDs at −150 mV in both monomeric and tetrameric

configurations. Since FRET efficiencies differed at hyperpolarized voltages (NMDG and resting) but were mostly similar at depolarized voltages (high KCl) among various hERG mutations, we performed unbiased simulations of the relaxed VSD of the R531Q mutant at a hyperpolarized voltage (−150 mV) and in a lipid environment to identify potential structural differences compared to the WT (Fig. 4a). These simulations were also started using homology models of hERG based on the cryo-EM structure of Eag channels with VSD in the down state[12]. To simplify the analysis, we modeled full-length hERG tetramers as well as hERG monomers including only the S1–S6 helices, incorporating R531Q. We found that in both models, the S4 helix in VSDs containing the R531Q mutation was elevated relative to the WT, stabilized in a position intermediate between the intermediate I and the S4-up conformations (Fig. 4a). Perhaps for the WT, the S4 may transition to the up position more readily, without encountering the energetic barrier that appears to trap R531Q in an intermediate state. This structural shift caused by R531Q is consistent with the reduced tmFRET efficiency and the corresponding FRET-predicted increase in donor–acceptor distances observed in the presence of R531Q (Fig. 4b and see details in Methods). Additionally, at hyperpolarized −150 mV, R531Q caused a nearly 30° tilt in the lower half of the S4 (Fig. 4a and Supplementary Fig. 9). R531Q also altered the secondary structure of the S4, switching it from a predominantly α-helical configuration in the WT to a more pronounced 3$_{10}$-helical conformation (Supplementary Fig. 9a, b).

To explore the dynamic pathway of the S4 movement and its alteration by the R531Q mutation, we performed nonequilibrium MD simulations, specifically steered MD (SMD). The 100 ns SMD started from the intermediate conformation I to the up-state conformation of the S4, in a monomer transmembrane domain (S1-S6 helices) under 0 mV (Fig. 4c, e–g, and Supplementary Movie 1). As an SMD reference, we used the central positions of the K525, R528, and R531 side chains relative to the S2 helix as collective variables[57–59]. The S4 showed the expected upward displacement of the S4, adopting a distinct intermediate conformation, termed intermediate II, likely driven by competition between R528 and K525 for interaction with D456 in the S2 helix (Fig. 4c, e–g). The S4-up position was subsequently stabilized by the interaction between R531 and D460 in the S2 helix. In this context, R531 promotes S4 upward movement, while K525 (K1) acts as an impediment, helping to explain why the LQTS-associated K525N mutation enhances hERG channel sensitivity to depolarizing voltages[25,60]. Overall, the S4 helix exhibited a modest displacement, with a 5 Å shift in the R531 side chain (Fig. 4f) and a 3 Å shift in the backbone of its N-terminal region (Fig. 4g), primarily reflecting the movement of R531 from below to above F463 in the charge transfer center. The modest scale of S4 movement is consistent with the relatively shallow G–V relationship of hERG channel activation. Note that the full range of S4 displacement is likely greater, as our SMD simulations begin from the intermediate I state rather than the fully down state.

In SMD simulations of the R531Q mutant, the S4 helix began in a more elevated position, resulting in subtler displacement during the first half of the simulation (Fig. 4d, e–g, and Supplementary Movie 2). No distinct intermediate dwell states were observed between the initial (intermediate I) and up conformations. These altered dynamics in the R531Q mutant likely contribute to its impaired voltage-sensor activation observed in functional studies.

## The R528Q mutation promotes similar FRET intermediates and S4 rearrangement

To further elucidate the mechanisms by which arginine mutations impact S4 helix movement, we tested the presence of intermediate FRET states in hERG channels containing the R528Q mutation using the same tmFRET design. The R528P mutation is a recently identified mutation associated with LQTS[27]. However, because proline is a helix-

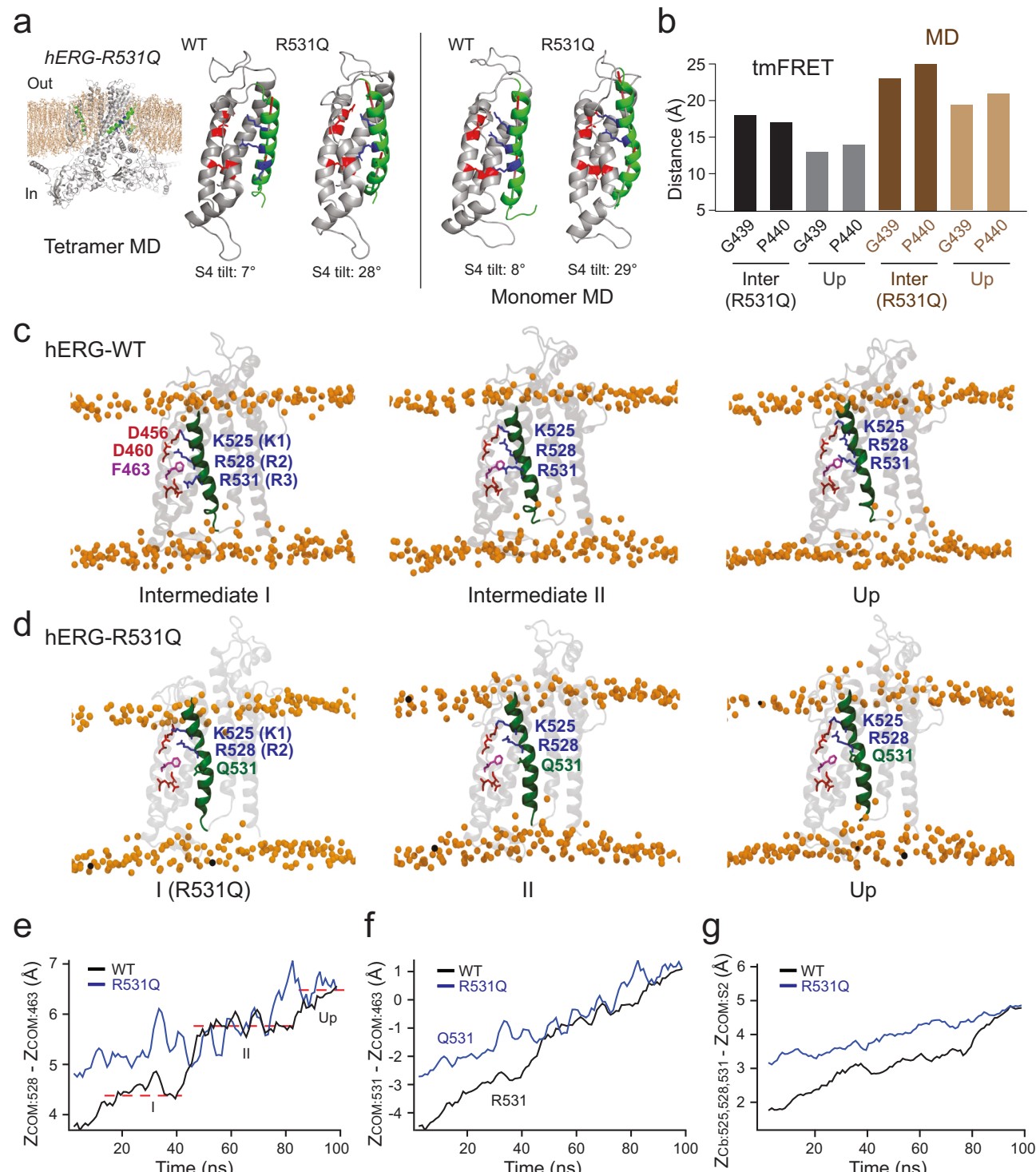

**Fig. 4 | R531Q induces displacement and increased tilt of the S4 helix in MD simulations. a** MD simulation of the full-length hERG-R531Q tetramer showing the S4 helix in an elevated and tilted conformation compared to the hERG-WT tetramer. A close-up view of the S4 helix is shown for one representative subunit. Similar R531Q-induced conformational changes in the S4 helix were observed in simulations of hERG transmembrane-domain monomers. The helix angels were estimated using the PyMOL plugin AngleBetweenHelices. **b** Comparison of tmFRET-estimated distances in hERG$_{tmFRET}$-R531Q channels between the intermediate FRET state and the high FRET state for two FRET pairs (G439/L523 and P440/L523). These distances were compared to Cα–Cα distances of the corresponding residue pairs derived from MD simulation-based homology models. Representative snapshots of hERG-WT-VSD conformations (**c**) and hERG-R531Q-VSD conformations (**d**) captured along the trajectory of steered MD-induced S4 rearrangement. Membrane lipid head groups are depicted as yellow spheres. **e** Time course of the distance of the center of mass (COM) of the R528 side chain in the WT and the R531Q mutant, measured relative to the COM of a reference residue, F463, during the SMD simulation. **f** Time course of the distance between COM of the Q531 side chain and COM of the F463 side chain in the R531Q mutant, along with the corresponding distance in the WT, over the course of the SMD trajectory. **g** Time course of the average combined distance between the Cβ atoms of residues 525, 528, and 531 in the R528Q mutant and the WT, measured relative to the COM of the S2 helix during the SMD simulation.

breaking residue and may cause protein misfolding, we did not include the R528P in this study.

As mentioned earlier, R528Q mutation induced biphasic channel activation, yet did not significantly alter the G-V relationship compared to WT+ channels (Supplementary Fig. 1d). Similar to the tmFRET results of R531Q mutant channels, phasor analysis revealed the presence of at least two different FRET efficiencies: a lower one around 0.35 (a converted distance of 16.6 Å) and a higher one around 0.5 (Fig. 5a–d), suggesting that the intermediate state of S4 with R528Q likely resides in a similarly more elevated position as in the R531Q mutant, and is thus designated as intermediate state II. Notably, unlike the R531Q mutant channels (as depicted in Fig. 2g, Fig. 3g compared with Fig. 5d), membrane depolarization increased the proportion of FRET states for R528Q-containing channels, indicating a higher probability of the S4 helix in the intermediate II or the up states. This observation contrasts with the behavior observed in R531Q channels and is consistent with the relatively unaffected $V_{1/2}$ in the G–V relationship of R528Q channels (Supplementary Fig. 1d).

In MD simulations of the hERG-VSD containing the R528Q mutation, we observed a similar elevation and tilt of the S4 segment at the resting hyperpolarized voltage (−150 mV) (Fig. 5e–h and Supplementary Fig. 9). The S4 tilting, measured at 34°, occurs near the Q528 site, in contrast to the R531Q mutant, where the tilt is closer to the Q531 site (Supplementary Fig. 9c, e). R528Q produced a similar effect in promoting the $3_{10}$-helical conformation in this resting state. SMD simulations of the R528Q mutant suggest that prolonged dwell time in the elevated S4 position may underlie the intermediate FRET observed (Fig. 5e–h, and Supplementary Movie 3). Overall, S4 movement is less disrupted by the R528Q mutation than by R531Q, consistent with the markedly greater functional impact of R531Q compared to R528Q or other S4-arginine neutralizations.

Additionally, we performed analogous equilibrium MD simulations of the R534Q and K525N mutants at −150 mV. These charge-neutralizing mutations also induced minor tilting of the S4, though the structural changes caused by R534Q and K525N were considerably less pronounced compared to those in R528Q and R531Q mutants (Supplementary Fig. 9). To assess functional consequences, we carried out parallel tmFRET experiments on hERG$_{tmFRET}$ channels including the LQTS-associated R534Q mutation (Supplementary Fig. 10). These experiments revealed a single high FRET state, distinct from the profiles of R528Q and R531Q mutants, but similar to hERG$_{tmFRET}$ channels, suggesting a different disease-causing mechanism that affects S4 movement. R534 is in the lower portion of the S4 helix and may reside outside or only partially within the transmembrane electric field during voltage sensing. The inactivation of R534L (and R534Q) at more depolarized voltages may instead underlie its contribution to LQTS (Supplementary Fig. 1e). Furthermore, analogous experiments using hERG$_{tmFRET}$ channels with the K525N mutation revealed also a single high-FRET state, but with a substantially higher FRET efficiency of 62 ± 0.5% ($n$ = 12 cells). This result is consistent with the idea that the S4 can move up further when the hindrance of K525 is abolished, which facilitates channel opening[25,60].

### Free-energy landscape of hERG voltage sensor dynamics
Dwells and pauses in the S4 displacement time course during SMD simulations suggest the presence of intermediate conformational states. To further assess the stability and energetic profiles of these putative metastable intermediates, we performed umbrella sampling simulations (Fig. 6)[57,58]. In this approach, multiple simulations were conducted with the system restrained at defined values of collective variables (CVs), referred to as windows along a reaction coordinate describing relative displacement of S4, without external voltage (0 mV). One-dimensional potentials of mean force (PMFs) were then calculated for the WT, R531Q, and R528Q VSDs. PMFs along the reaction coordinate were derived by reconstructing free energy profiles

across all windows using the generalized weighted histogram analysis method (GWHAM)[61]. This analysis provided detailed energetic insights into how specific mutations reshape the conformational landscape of the S4. For the WT VSD, the PMF revealed at least two distinct free energy minima, corresponding to intermediate I and the up state (Fig. 6a). Importantly, in the WT free energy landscape, the final step appears to confer a kinetic barrier that hinders reversal (Fig. 6a). Furthermore, the R531Q and R528Q mutants also exhibited noticeable free energy minima corresponding to the SMD-observed dwells, indicating these as stable intermediate states (Fig. 6 b, c). In contrast to the WT, the R531Q mutant exhibits a smoother energy landscape with less barriers, allowing the S4 to transition more freely, including reversals to the more stable intermediate I state. This suggests a less stabilized up state (Fig. 6b). The R528Q mutant, by comparison, displays a free energy profile with multiple stable intermediates, including a lowest-energy intermediate II state (Fig. 6c). This altered energy landscape may underlie the characteristic slow deactivation, multi-exponential activation kinetics, as well as the observed intermediate II FRET state of the R528Q mutant.

Together, our findings suggest that mutating key voltage-sensing arginines in hERG channels stabilize intermediate conformational states of the S4 helix. These intermediates provide a structural framework for interpreting and comparing to analogous states identified in voltage sensors of other VGICs[12,62–64]. Integration of electrophysiology, FLIM-tmFRET, and MD simulations provides a comprehensive understanding of the mechanisms by which LQTS-associated mutations impair voltage sensing in hERG channels.

## Discussion
Currently, effective methods for probing the conformational dynamics and heterogeneity of membrane proteins in living cells are limited. While single-molecule FRET techniques offer substantial advantage in dissecting protein conformational states, their application in studying rapid and voltage-dependent events within native membranes remains challenging. Similarly, obstacles persist in employing methods such as cryo-EM and double electron-electron resonance (DEER), both of which routinely require cryogenic sample freezing. Fluorescence lifetime decays, like DEER signal decays, contain information that can be analyzed through multi-exponential functions, potentially unraveling distinct populations of fluorophores. Notably, these fluorescence lifetime techniques can be executed within native membranes and integrated with functional approaches like patch clamp. Leveraging a combination of genetic-code expansion, click chemistry, frequency-domain phasor FLIM, and tmFRET, we engineered a live-cell system to measure intermediate states of ion channel voltage sensors in a relatively high-throughput manner. While the phasor FLIM technique has been used in quantifying multiple states of FRET in the field of cell biology[22], its potential in elucidating the structural biology of proteins is only beginning to be realized[23,53]. Our integrated approach offers a broadly applicable framework for studying protein dynamics, particularly for membrane proteins.

Building upon our findings, we postulate that neutralization of S4 positive charges in the charge transfer center (R2 and R3) gives rise to distinctive intermediate states within the S4 helix voltage sensor (Fig. 6d). Specifically, the negatively charged residues situated within the S2 and S3 helices, known as the extracellular negative cluster (ENC; D456 and D460) and intracellular negative cluster (INC; D411, D466 and D501), play a role in stabilizing these intermediate states. In the case of R531Q mutation, the positively charged side chain of R528 is essential for S4 to interact with the ENC, thereby stabilizing its intermediate state. In contrast, in the instance of the R528Q mutation, S4 depends on the positive charge of the lower R531 to interact with the ENC, thereby resulting in a slightly upward shift of S4, compared to that for the R531Q mutation (Fig. 6d). In other words, as the S4 moves upward, the intermediate states get stabilized when neutralized

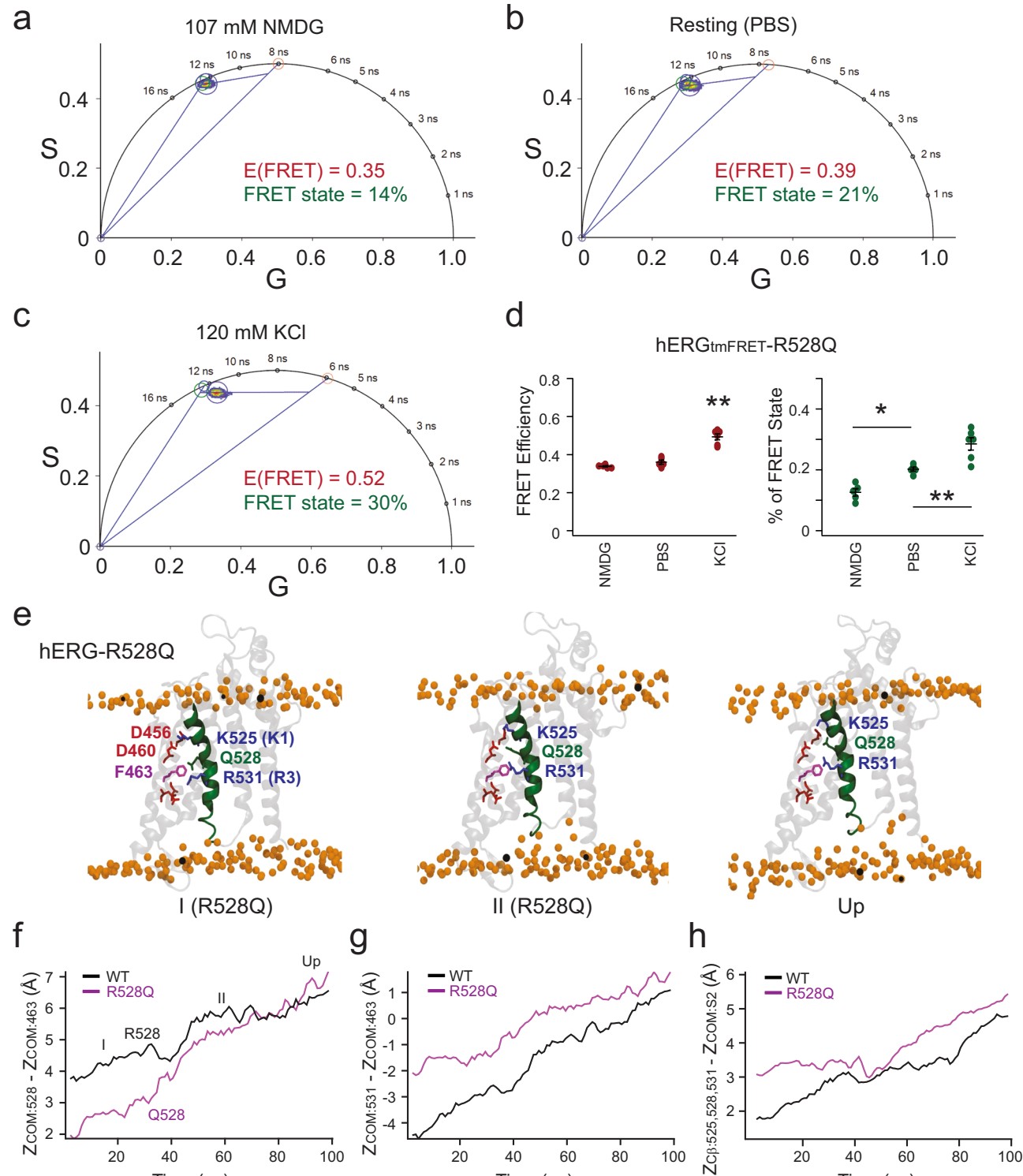

**Fig. 5 | Distinct S4 conformations induced by R528Q suggested by FLIM-tmFRET and MD simulations. a–c** Representative analysis of the hERG from individual cells, when R528Q was included. **d** Summary data of the estimated FRET efficiencies and percentage of the FRET state using linear unmixing of the phasor approach. Data shown are presented as mean ± s.e.m., Sample sizes: NMDG ($n = 5$ cells), dPBS ($n = 5$ cells), KCl ($n = 6$ cells). For the FRET efficiency, $p = 1e\text{-}6$ between NMDG and KCl conditions, and $p = 8e\text{-}6$ between PBS and KCl conditions, one-way ANOVA (no adjustment). For the percentage of FRET state, $p = 0.01$ between NMDG and dPBS conditions, $p = 1.4e\text{-}5$ between NMDG and KCl conditions, $p = 0.005$ between dPBS and KCl conditions. Source data are provided as a Source Data file. **e** Representative snapshots of hERG-R528Q-VSD conformations captured along the

trajectory of steered MD-induced S4 rearrangement. Membrane lipid head groups are depicted as yellow spheres. **f** Time course of the distance of the center of mass (COM) of the Q528 side chain in the R528Q mutant and COM of the R528 side chain in the WT, measured relative to the COM of a reference residue, F463, during the SMD simulation. **g** Time course of the distance between COM of the R531 side chain and COM of the F463 side chain in the R528Q mutant, along with the corresponding distance in the WT, over the course of the SMD trajectory. **h** Time course of the average combined distance between the Cβ atoms of residues 525, 528, and 531 in the R528Q mutant and the WT, measured relative to the COM of the S2 helix, during the SMD simulation.

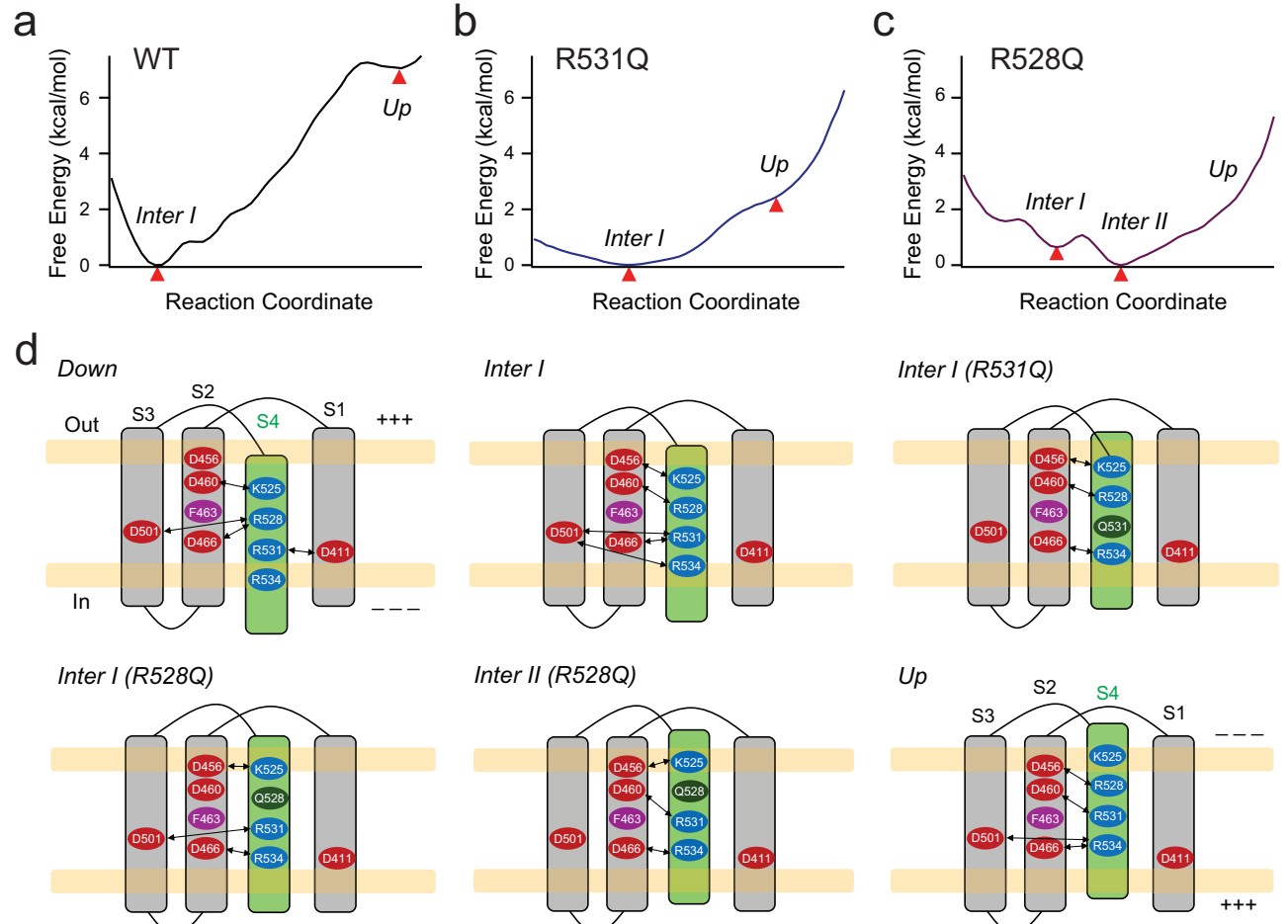

**Fig. 6 | Conformational free energy landscape of S4 movement in wild-type, R531Q, and R528Q hERG channels.** One-dimensional potential of mean force (PMF) profiles for the monomer transmembrane domains of wild-type WT (**a**), R531Q (**b**), and R528Q (**c**) hERG variants, obtained via umbrella sampling based on steered molecular dynamics trajectories along a defined reaction coordinate. The profiles depict free energy minima along the collective displacement of residues 525, 528, and 531 (used as collective variables) as the S4 transitions from the intermediate I to the up conformation. Free-energy minima are highlighted by triangles in red. **d** Schematic representations of the VSD topology and associated charge interactions in the down, intermediate I, intermediate II, and up states. Negatively charged residues (in red) within the S1 to S3 helices and positively charged residues (in blue) within the S4 are highlighted. The intermediate I conformation differs between the WT and the R531Q or R528Q mutants. The intermediate II is most stable in the R528Q mutant.

arginine side chains fail to exchange their ion pair partners[64]. Interestingly, the contrasting activation patterns observed between R531Q channels (long initial lag and mono-exponential activation) versus R528Q channels (short initial lag and double-exponential activation) may be due to the distinctive functional coupling of the intermediate II S4 conformation to channel opening. Stabilizing the intermediate II conformation may hinder pore closure, as indicated by the significantly slower deactivation observed in the R528Q mutant, which favors a low-energy intermediate II state in its free energy landscape (Fig. 6c). Conversely, stabilizing the intermediate I conformation in R531Q could prevent pore opening (Fig. 6b). Furthermore, disrupting the characteristic "RXX" repeat can split the movement of the voltage sensor into distinct phases, a behavior reminiscent of the biphasic S4 movement observed in the VSD of various other proteins and channels[63,65–67]. Finally, our results provide insights to analogous S4-arginine (R2 and R3 positions, Supplementary Fig 1) mutations that cause cardiac diseases in other types of VGICs, e.g., in voltage-gated Na+ channels[68–71].

Our MD simulations provide insights into S4 movement in VGICs, particularly those in non-domain-swapped subunit arrangements, such as KCNH family channels[72,73]. They built upon structural data of the Eag channel in the S4 down-state (cryo-EM)[12], and employed extended, structurally guided protocols that enable more accurate modeling of S4 conformations at hyperpolarized voltages and the dynamic movement of S4—an advance over earlier methods[74]. The elevated S4 helix position induced by arginine mutations aligns with intermediate states predicted by FRET experiments. While some S4 arginine mutations generate gating pore currents in VGICs[75,76], R531Q produced little ionic current at hyperpolarized voltages. In non-domain-swapped VGICs, the closer spatial proximity of S4 to S5 raises the possibility that arginine mutations may influence the pore domain through distinct mechanisms compared to domain-swapped VGICs[9,11,12,73]. Future studies should explore how these structural rearrangements induced by mutations modulate VSD-PD coupling within the context of tetrameric channels. Additionally, it remains unclear whether a potential $3_{10}$-helix to α-helix conversion modulates the intermediate state of voltage sensors[12,16,25,77,78]. In our SMD (Supplementary Movies 1–3), we noticed that during the upward movement of S4, S4 regions transitioned from an α-helical to a $3_{10}$-helical conformation while traversing the charge transfer center. This possible rearrangement allows the voltage-sensing arginines to align on one side of the helix with minimal rotation of S4[12]. The observation that arginine mutations promote a $3_{10}$-helical conformation at the resting S4 (intermediate I) state suggests that such structural alterations could contribute to the functional changes, potentially by affecting the stability, movement, or energetics of the S4 in VGICs. Moreover, the tilt of

the S4 caused by arginine mutations is reminiscent of similar motions observed in the S4 of non-domain-swapped VGICs, particularly in HCN channels[10,18,50,79]. This suggests that the C-terminal half of the S4 segment in this VGIC configuration is highly susceptible to tilting or bending motions, which plays a significant role in shaping the diversity of voltage-dependent gating mechanisms in KCNH channels.

Our fluorescence method represents a significant advancement in applying FLIM-FRET to structural biology in live-cell imaging and holds potential for further refinement and development. Enhancing the selectivity of pyrrolysyl-tRNA synthetase for Acd is essential to minimize nonspecific fluorescence. The presence of endogenous genes using the amber codon might interfere with the Acd system, resulting in undesired cytosolic Acd fluorescence in our preparations. Developing an Acd variant that fluoresces exclusively upon incorporation into the protein of interest would help mitigate the nonspecific Acd fluorescence as well. Furthermore, the inherent flexibility of the extended linker length of the ProK-Azide-DOTA remain limiting factors for precise absolute distance measurements. Optimizing the linker length between ProK and azido DOTA could bring the transition metal acceptor closer to the protein backbone. This engineering can also benefit from more efficient CuAAC click chemistry, requiring lower concentrations of Cu(I) catalyst and exhibiting faster reaction kinetics. Alternatively, engineering copper-free strain-promoted azide-alkyne cycloaddition (SPAAC) reactions could also facilitate labeling of the tmFRET acceptor[80,81]. Such improvements would not only enhance the accuracy of this method but also reduce the cytotoxicity associated with transition metals. Finally, incorporating patch clamp techniques into this method to precisely control membrane voltage could further enhance our ability to correlate channel function with structural states.

## Methods
### Molecular cloning
Bioorthogonal labeling of noncanonical amino acids (ncAAs) was achieved by site-specific incorporation using either single or dual codon suppression. This involved using amber (TAG) and ochre (TAA) codons within the channel protein of interest, specifically the hERG1 cDNAs synthesized by VectorBuilder (Chicago, IL). These constructs, featuring a C-terminal eYFP and a CMV promoter, were cloned and amplified using a mammalian gene expression vector. The S620T mutation, located in the pore helix of the channel, was included in all hERG1 constructs used in this study to eliminate rapid C-type inactivation[30–32]. This modification allows for a more accurate estimation of voltage-dependent activation by avoiding the overlap between activation and inactivation phases. Moreover, we believe that the S620T mutation does not impact the structure and function of the VSD. For simplicity, we refer to the modified hERG1-S620T-YFP channel as the "WT+" channel, and the hERG1-L523TAG/G439TAA, S620T-YFP as "hERG$_{tmFRET}$" throughout this paper.

DNA oligo primers, from Integrated DNA Technologies (IDT, Coralville, IA), were used for the design and synthesis process (Supplementary Table 2). Point mutations were induced using either the QuikChange II XL Site-Directed Mutagenesis or Multi Site-Directed Mutagenesis Kit (Agilent Technologies, Cedar Creek, TX). Subsequently, Dpn I restriction enzyme treatment (Agilent Technologies) was applied to digest the methylated parental DNA template of the DNA plasmids, which were then transformed into XL10-Gold competent cells (Agilent Technologies). The introduction of double functional stop codons opal (TGATGA) at the terminal end of YFP ensured successful translation termination. Verification of the introduced stop codons within the mutant plasmid was performed through DNA sequencing provided by either Genewiz (Azenta Life Sciences, Burlington, MA) or Plasmidsaurus (Eugene, OR). Additionally, the DNA concentration was quantified using a Nanodrop OneC spectrophotometer (Life Technologies, Grand Island, NY).

### Cell culture, transient transfection and bioorthogonal labeling
The tsA-201 cells (catalog number: 96121229), a variant of human embryonic kidney (HEK293) cells, were procured from Sigma-Aldrich (St. Louis, MO), authenticated via short tandem repeat (STR) profiling, and then cryopreserved in liquid nitrogen. For initiating cell cultures, these cells were thawed and cultured in Dulbecco's modified Eagle's medium (DMEM; high glucose, pyruvate, 11995, Gibco), supplemented with 10% fetal bovine serum (FBS; Gibco) and 1% penicillin/streptomycin (Gibco), within a humidified incubator maintained at 37 °C with 5% $CO_2$.

Transfection was performed on 70%–90% confluent tsA-201 cells using the lipofectamine 3000 kit (Invitrogen, Carlsbad, CA, #L30000080), following established protocols[82,83]. For dual stop-codon suppression, the cells were co-transfected with two plasmids: pAS_4xU6-PyIT M15(UUA) FLAG-Mma PyIRS (Mma PyIT/RS) from Addgene (Watertown, MA, #154774)[41] and the Acd RS1 tRNA/aminoacyl-tRNA synthetase pair (GCE4All Research Center, Oregon State University, and Addgene #197566)[40], along with a third plasmid containing the dual stop-codon mutations (TAG and TAA) of the hERG gene required for selective incorporation of two noncanonical amino acids (ncAAs): Acd (the Petersson Lab, University of Pennsylvania) and N-propargyl-L-lysine (ProK) (Catalog #SC-8002, SiChem, Bremen, Germany)[41]. This Acd RS1 tRNA/aminoacyl-tRNA synthetase pair consists of one copy of aminoacyl-tRNA synthetase and four copies of tRNA. To facilitate nonsense suppression, a human dominant-negative release factor eRF1 (E55D) (VectorBuilder, Chicago, IL) was co-transfected[84]. The ncAAs were added to the transfection solution: 5 μM Acd and 0.25 mM ProK. ProK was mixed with 1 M HEPES solution in a 1:4 ratio (Sigma-Aldrich, St. Louis, MO) before adding to cell culture. The ProK stock solution (100 mM) were made by dissolving ProK in 0.2 M NaOH water. In the absence of the pAS_4xU6-PyIT M15(UUA) FLAG-Mma PyIRS (Mma PyIT/RS) while in the presence of the Acd incorporation system, we did not observe significant membrane-localized YFP signals (Supplementary Fig. 2f), indicating the Acd RS1 tRNA/aminoacyl-tRNA synthetase pair is specific to the amber stop codon and cannot suppress the ochre stop codon. In addition, we conducted experiments expressing hERG$_{tmFRET}$ constructs with aminoacyl-tRNA synthetases but without Acd and ProK amino acids and observed no detectable currents—similar to untransfected cells—in the presence of 10 mM TEA (T2265, Sigma-Aldrich) in the bath solution.

At the ProK site, we conducted the copper-catalyzed azide-alkyne cycloaddition (CuAAC) reaction using 50 μM CuSO$_4$, 250 μM THPTA (Tris(benzyltriazolylmethyl)amine) (762342, Sigma-Aldrich), and 2.5 mM ascorbic acid (A4544, Sigma-Aldrich) in DMEM medium[41]. For azido-DOTA labeling (at 100 μM in cell culture), the incubation proceeded for 1–1.5 h at 37 °C. To produce Cu$^{2+}$-DOTA, we mixed equal amounts of 100 mM azide DOTA stock and 110 mM CuSO$_4$ stock, allowing them to incubate for five minutes until the mixture transitioned from light to darker blue, indicating Cu$^{2+}$ binding to DOTA. Once bound, DOTA and Cu$^{2+}$ dissociation occurred very slowly due to their picomolar affinity[38]. Azido-mono-amide-DOTA (1,4,7,10-Tetraazacyclododecane-1,4,7-tris-acetic acid-10-azidopropylethylacetamide, product #B-288) was purchased from Macrocyclics, Inc (Plano, TX). We tested different durations for the azido-Cu$^{2+}$-DOTA labeling (30 mins, 1 hour, and 1.5 h) to assess completeness and found that the Acd lifetime shift plateaued around 1–1.5 h, which is consistent with the relative fast kinetics for this type of click chemistry[80]. For certain acceptor sites such as A430, which is more membrane-embedded, slightly longer labeling times may be required to achieve full labeling efficiency. However, azido-Cu$^{2+}$-DOTA labeling beyond 2 h caused noticeable cytotoxicity; therefore, labeling duration was restricted to a maximum of 2 h.

## Fluorescence microscopy, fluorescence lifetime measurement and FLIM-FRET

Fluorescence measurements were conducted using an Olympus IX73 inverted microscope (Olympus America, Waltham, MA), equipped with a 60× U Plan S-Apo 1.2-NA water-immersion objective. Epifluorescence recording was facilitated by wide-field excitation from a pE-800 LED light source (CoolLED, Andover, UK). For choosing cells with strong membrane localized YFP fluorescence via eyepiece of the microscope, YFP was excited using a 500 nm LED line. Acd fluorescence was expected in these cells and was excited using a 365 nm LED line. The imaging solution used for cell visualization was Dulbecco's phosphate-buffered saline (PBS, ThermoFisher), unless specified otherwise. All microscopic experiments were conducted at room temperature.

Spectral measurements of Acd fluorescence used a spectrograph (SpectroPro 2156, 300 groves/mm grating, 500 nm blaze; Teledyne Princeton Instruments, Acton, MA) situated between the microscope output port and an INFINITY3S-1URM CCD camera (Teledyne Lumenera, Ottawa, ON). Acd fluorescence from cells was excited using a 365 nm LED line through a filter cube. The emission filter was removed from the filter cube, allowing only fluorescence in the wavelength range of 423 to 597 nm collected as a spectral image. Spectral images were acquired with exposure times of 500 ms. In addition, spectral measurements of the light absorption of transition metals were made using the Nanodrop OneC spectrophotometer (Life Technologies, Grand Island, NY). Subsequent analysis of the spectral data was conducted using ImageJ/Fiji (National Institutes of Health, USA) and Igor Pro 9 (Wavemetrics, Portland, OR).

Digital frequency-domain fluorescence lifetime imaging was conducted using a Q2 laser scanning confocal system, which was equipped with a FastFLIM data acquisition module (ISS, Inc, Champaign, IL) and two hybrid PMT detectors (Model R10467-40, Hamamatsu USA, Bridgewater, NJ)[21,38,85]. This confocal FLIM system is integrated with the Olympus IX73 inverted microscope. The fluorescence lifetime measurements encompass both frequency-domain (allowing instantaneous distortion-free phasor plotting) and time-domain decay information. To excite Acd, a 375 nm modulated diode laser (ISS, Inc) controlled by FastFLIM was used, with the emitted fluorescence captured through a 447/60 nm band-pass emission filter. YFP excitation was achieved using a 488 nm wavelength excitation and collected through a 542/27 nm emission filter. Simultaneous two-color imaging was facilitated by dichroic cubes, with one containing a 50/50 beam splitter and the other equipped with long-pass filters.

For image acquisition, confocal images with a frame size of 256 × 256 pixels were obtained for each experimental condition, using a motorized variable pinhole (set at a size of 100 μm). To determine the phase delays ($\varphi$) and modulation ratios ($m$) of the fluorescence signal in response to an oscillatory stimulus (6 ps per pulse) with a frequency $\omega = 20$ MHz, the VistaVision software (ISS, Inc) associated with the FastFLIM system was used. These values were obtained by performing sine and cosine Fourier transforms of the phase histogram, taking into account the instrument response function (IRF) calibrated with Atto 425 in water (with a lifetime of ~3.6 ns, see Supplementary Fig. 2). VistaVision software, equipped with the imaging and FLIM module, was also used for image processing, display, and acquisition. This software allowed for the specification of parameters such as pixel dwell time, region of interest, and resolution. Additionally, it employed a fitting algorithm, phasor linear unmixing analysis for FLIM-tmFRET analysis. To enhance display and analysis of membrane-localized lifetime species, a median/Gaussian smoothing filter and an intensity-based selection of the membrane region were applied. The VistaVision and phasor FLIM provide a significant advantage by effectively differentiating signals originating from background and cytosolic fluorescence outside the membrane region from those within the plasma membrane. This capability is especially important given that our Acd

incorporation system tends to generate substantial nonspecific cytosolic Acd fluorescence, which constitutes the primary error source in our experiments. Relying on the selection of cells exhibiting strong membrane-localized YFP fluorescence enabled us to analyze lifetime species of the Acd fluorescence mostly aligned with YFP fluorescence associated the plasma membrane (Supplementary Fig. 2). The exact reason for the greater cytosolic Acd fluorescence comparing to other systems that incorporate fluorescent noncanonical amino acids (e.g. L-Anap) via stop-codon suppression is unclear[38,40,45,86], and likely due to the high membrane permeability of Acd, potential high affinity between Acd and its tRNA, and nonspecific amber stop-codon suppression of endogenous genes. Additionally, we engineered a similar plasmid containing the Acd RS1 tRNA/aminoacyl-tRNA synthetase pair with one copy of aminoacyl-tRNA synthetase but only two copies of tRNA, which was sufficient to achieve dual-stop-codon suppression, indicated by significant membrane-localized YFP fluorescence. However, this modification did not reduce the nonspecific cytosolic Acd fluorescence.

The VistaVision software features the FRET trajectory function to estimate the FRET efficiency, manifesting an averaged distance between donor and acceptor probes. Key parameters are adjusted to optimize the fitting trajectory through both the donor alone and the donor species due to quenching by FRET acceptors[38]. Regarding the linear unmixing (decomposition) approach of calculating FRET efficiency, the linear fitting process underwent visual inspection to derive four parameters: (1) background contribution in the donor-alone (or low FRET) sample, (2) background contribution in the donor sample in the presence of the acceptor, (3) percentage contribution by the FRET state, and (4) FRET efficiency under theoretical conditions where 100% donors undergo high FRET. Background fluorescence remained within the cytosol and the levels typically at or below 5%, consistent with the minimal emission intensity observed in untransfected cells. To determine the percentage contribution by the FRET state, the cursor position was adjusted to cover the central portion of the lifetime distribution. The FRET efficiency of the low FRET state corresponding to the down state of the S4 cannot be accurately measured since the lifetime species typically would not exhibit an elliptical distribution and localize very close to the donor-alone phasor.

The distances $r$ between the Acd and Cu$^{2+}$-DOTA acceptors were estimated using the convolution of the Förster equation ($E_{max.} = 1/(1 + (r/R_0)^6)$) with a Gaussian function: Förster convolved Gaussian (FCG) as previously established[10,86]. The FCG adopts a Gaussian distribution with a standard deviation ($\sigma$) of 7.5 Å and a full width at half maximum (FWHM) of 17.7 Å (Supplementary Fig. 5). This assumption was based on the interatomic distance distribution of probes with similar labels, e.g., the L-Anap and Cu$^{2+}$-TETAC tmFRET pair[10,24,86]. The $R_0$ value was calculated using the following equation:

$$R_0 = C\sqrt[6]{J Q \eta^{-4} K^2} \tag{1}$$

where $C$ is the scaling factor, $J$ is the overlap integral of the donor emission spectrum and the acceptor absorption spectrum, $Q$ is the quantum yield of the Acd (~0.8), $\eta$ is the index of refraction, and $\kappa^2$ is the orientation factor. $\eta$ was assumed to be 1.33, and $\kappa^2$ was assumed to be 2/3. This calculation generated a $R_0$ of 13 Å between Acd and Cu$^{2+}$-DOTA and 12 Å between Acd and Co$^{2+}$-DOTA.

To assess the effectiveness of the FCG-based distance estimation, we compared the distances estimated by tmFRET for the L523/G439 and L523/P440 FRET pairs to the corresponding Cβ(L523)−Cα(G439 or P440) distances from MD simulations of the R531Q VSD at hyperpolarized voltages and the WT VSD in the up state (Fig. 4b). While the absolute distances predicted by tmFRET were about 5 Å different from the Cβ−Cα distances—likely due to the linker lengths and side-chain positions of the FRET donors and acceptors—the change in distance ($\Delta D$) from the intermediate state of R531Q to the up state closely

agreed between the two approaches. For the L523/G439 pair, the $\Delta D$ was 5 Å from tmFRET and 4 Å from MD simulations; for the L523/P440 pair, the $\Delta D$ was 3 Å and 4 Å, respectively (Fig. 4b). These findings underscore the accuracy of our phasor FLIM-tmFRET method in predicting conformational changes within the VSD.

## Patch-clamp electrophysiology

Whole-cell patch-clamp recordings were conducted at a temperature range of 22–24 °C using an EPC10 patch-clamp amplifier and PATCH-MASTER (HEKA) software. Borosilicate patch electrodes, with an initial pipette resistance of approximately 3–5 MΩ, were fabricated using a P1000 micropipette puller from Sutter Instrument (Sutter, Novato, CA). Current signals were low-pass filtered and sampled at a frequency of 10 or 5 kHz. Series resistance was mostly under 10 MΩ and compensated at ~40–70%. Patch clamp was performed using a Sutter MP-225A motorized micromanipulator (Sutter, Novato, CA).

To record hERG currents from tsA-201 cells, the internal pipette solution was prepared with the following composition (in mM): 10 NaCl, 130 KCl, 3 MgCl₂, 10 HEPES, 0.5 EGTA, 1 MgATP, 0.3 Na₃GTP, and pH 7.2 adjusted with KOH. The external solution used was Ringer's solution with the following composition (in mM): 160 NaCl, 2.5 KCl, 1 MgCl₂, 2 CaCl₂, 10 HEPES, 8 D-Glucose, and pH 7.4 adjusted with NaOH. 10 mM TEA was added to the external solution to block endogenous potassium currents of tsA cells, while having minor effect on the hERG currents[29]. For hyperpolarizing and depolarizing solutions, 107 mM impermeant cation NMDG (N-methyl-D-glucamine) and 120 mM KCl replaced NaCl in the Ringer's solution, respectively[48].

The conductance-voltage (G–V) relationships were determined by analyzing the instantaneous tail currents subsequent to voltage pulse steps. Subtraction of leak tail currents was performed, and the resulting currents were normalized to the maximum tail current amplitude ($G/G_{max}$). The relative conductance was then plotted against the voltage of the main pulse and fitted using the Boltzmann equation:

$$G/G_{max} = 1/\left(1 + \exp\left[\left(V_{1/2} - V\right)/V_s\right]\right) \quad (2)$$

where $V$ is the membrane potential, $V_{1/2}$ is the potential for half-maximal activation, and $V_s$ is the slope factor. The difference in free energy between the channel closed state and open state at 0 mV was calculated according to: $\Delta G = RTV_{1/2}/V_s$, where $R$ is universal gas constant and $T$ is room temperature in kelvin (297 K). A more positive $\Delta G$ value indicates a greater energetic favorability for the channel to remain closed at 0 mV.

As a comparison, to calculate the Boltzmann free energy difference ($\Delta G$) between the channel opening and closing at the high KCl condition in our FRET experiments, given the fraction of S4 in the activated high-FRET state, we used the following equation derived from the Boltzmann distribution:

$$\Delta G = -RT\ln\left(\left(P_{FRET}\right)^n/\left(1 - P_{FRET}\right)^n\right) \quad (3)$$

where $P_{FRET}$ is the fraction of voltage sensors in the high FRET state. This expression assumes a tetrameric model, in which at least n of the 4 independent subunits need to be in the high FRET (activated) state for the channel to be considered open. We estimated the channel open probability ($P_{open}$) from $P_{FRET}$ using the following binomial equation:

$$P_{open} = \sum_{k=3}^{4} \binom{4}{k} \left(P_{FRET}\right)^k \left(1 - P_{FRET}\right)^{4-k} \quad (4)$$

For hERG$_{tmFRET}$, when $P_{FRET}$ is 0.35, the estimated $P_{open}$, calculated as the sum of binomial probabilities for at least three subunits being active, is about 13%. In contrast, for the hERG R531Q mutant with a $P_{FRET}$ of 0.2, the estimated open probability drops to just 2.7%. Estimates using this "3-subunit" model align with the moderate activation of WT

channels and the minimal or absent activation of the R531Q mutant near 0 mV—a voltage approximately achieved by extracellular 120 mM KCl in tsA cells[38] (Supplementary Table 1). Neither the "1-subunit" nor the "4-subunit" model produces channel opening profiles that align well with experimental observations (Supplementary Table 1). The "2-subunit" model, while closer, is likely unrealistic, as hERG channels probably do not reach 100% open probability even at saturating depolarizing voltages.

## Modeling hERG VSD structures

Our initial hERG-VSD down-state model was constructed through homology modeling, using the cryo-EM structure of the VSD in the down state of an Eag channel (PDB: 8EP1) as a template[12]. To model the up-state conformation of the hERG voltage-sensing domain (VSD), we compared the cryo-EM structure of hERG (PDB: 5VA1)[14], a homology model based on the up-state VSD of the Eag channel (PDB: 8EOW), and an AlphaFold-predicted model[87]. All three models exhibited strong overall structural similarity. The S1-S2 loop region—where our FRET acceptor sites are located—is unresolved in cryo-EM structures of hERG[14], and was further modeled using AlphaFold to estimate donor–acceptor distances for FRET analysis.

All MD simulations in this study were conducted using NAMD3 with a 2 fs timestep[88,89]. The systems were equilibrated under the NPT ensemble, maintaining the temperature at 310 K with the Langevin thermostat[90]. The Nosé–Hoover Langevin piston method was used to keep the pressure at 1 bar, and $\tau_p = 0.1$ ps for 50 ns[91,92]. The protonation states for amino acid were assigned using the VMD psfgen plugin (www.ks.uiuc.edu/Research/vmd/plugins/psfgen)[93]: Arg and Lys were protonated, Asp and Glu were deprotonated, and histidines (HSD) had single protonation on the δ-nitrogen. Particle mesh Ewald (PME)[94] summation was used for long-range electrostatic interactions, with a 12 Å cutoff, and a 10 Å switching distance was applied for van der Waals interactions. The simulations were carried out with the CHARMM36 force field[95,96]. Each system underwent CHARMM-GUI's standard multi-step equilibration protocol[97], during which positional restraints were gradually released over 2 ns prior to production runs.

The down-state homology model was embedded in a lipid bilayer mimicking the sarcolemma, composed of phosphatidylcholine (PC, 33%), phosphatidylserine (PS, 3.5%), phosphatidylethanolamine (PE, 17%), phosphatidylinositol 4,5-bisphosphate (PI(4,5)P₂, 3.5%), cholesterol (33%), and sphingomyelin (10%)[98]. The system was built using CHARMM-GUI Membrane Builder[97] and solvated in 0.15 M KCl using the TIP3P water model[99] within a 205 Å x 205 Å x 190 Å unit cell[95,96]; the final assembly consists of 720k atoms. After equilibration, the WT tetramer system was simulated under an applied potential of −150 mV for 1 μs. The mutant tetramer systems were initialized from the final frame of the WT system and simulated at −150 mV for 250 ns. Point mutations were generated using the VMD Mutator Plugin[93,100]. Based on our observations that both WT and mutant systems reach convergence within 250 ns, no significant changes were observed in root-mean-square deviation (RMSD) of backbone atoms—a measure of structural stability—between 250 ns and 1 μs for the WT, the mutant tetramers were simulated for ~250 ns (Supplementary Fig. 11).

The monomeric transmembrane domain (TMD; residues 405–667 of hERG) was derived from the corresponding region of one of the subunits of WT tetramer model. It was embedded in a membrane with the same lipid composition as used in the equilibrium simulations described above and solvated in 0.15 M KCl using TIP3P water; the resulting system size was 150 Å x 150 Å x 130 Å; the total system size was 255k atoms. The monomer TMD was simulated at −150 mV for 250 ns to obtain a relaxed structure resembling the intermediate I state. Similarly, the mutant monomer systems were initialized from the final frame of the WT monomer counterpart and simulated at −150 mV for 250 ns. For the WT, R531Q, and R528Q monomer systems, two or three independent replicas were performed. Convergence was

evaluated by tracking RMSD of backbone atoms across trajectories (Supplementary Fig. 11), with all systems demonstrating stable behavior consistent with convergence by the end of the simulations.

## Steered molecular dynamics (SMD) simulations

For the SMD simulations, we started with the end state of monomer TMD simulations as the starting point. Simulations of the TMD are computationally efficient, making them a practical choice for SMD simulations. SMD was then used to steer the S4 from the intermediate I to the up state at 0 mV using three "distance Z" collective variables; the three collective variables were defined as the vertical displacement from the center of mass (COM) of K525, R528, and R531 sidechain atoms, with respect to the COM for the S2 helix backbone atoms. The target value for these variables was defined with respect to the structure of hERG, in the VSD up state captured by cryo-EM (PDB: 5VA1)[14]. Specifically, for SMD of the WT TMD, the distance values were 7.6 Å for K525, 3.0 Å for R528, −5.4 Å for R531 in the starting model, to 11.4 Å for K525, 6.3 Å for R528, and 0.9 Å for R531 in the up state. The S1-S3 helices were restrained with 10 kcal/mol/Å² harmonic restraints. For the SMD simulations of the R531Q and R528Q mutants, identical target values for the collective variables corresponding to the up (end) state were used. Simultaneously, a pulling force was applied to each collective variable using a harmonic potential with a spring constant of 90 kcal/mol/Å². The SMD simulation was performed over 100 ns.

## Umbrella sampling analysis of hERG S4 movement

To quantify the free energy landscape associated with vertical translation of the S4 helix in the hERG VSD, we performed umbrella sampling simulations based on the SMD of WT, R528Q, R531Q VSDs[101]. The averaged center-of-mass (COM) z-coordinate of K525, R528 (or Q528) and R531 (or Q531) relative to the COM of S2 helix was chosen as the reaction coordinate, representing vertical displacement perpendicular to the membrane plane. The sampling range for each system was derived from its respective SMD trajectory. Twelve umbrella windows were constructed to evenly span the reaction coordinate. Initial configurations for each window were extracted from the corresponding SMD trajectories. Window spacing was ~0.35 Å for WT and R528Q, and ~0.25 Å for R531Q, adequately covering the expected conformational transition between the initial and up state. Each umbrella window was energy minimized and equilibrated, followed by a 10 ns production run under the NPT ensemble using NAMD3 with a 2 fs timestep. The restraint force constant was set at 8 kcal/mol/Å² for the production phase. A harmonic restraint was applied to a collective variable defined as the distance between the average COM of residues K525, R528 (or Q528), and R531 (or Q531) and the COM of S2 helix. The collective variable and restraint potential were implemented using the Colvars module in NAMD3[102,103]. The temperature was maintained at 310 K using the Langevin thermostat, and pressure was held at 1 atm using the Nosé–Hoover Langevin piston method. Long-range electrostatics were treated using PME, and van der Waals interactions were truncated with a switching function from 10 to 12 Å. The final 5 ns of trajectory data from each window were used to reconstruct the potential of mean force (PMF) profile using the Weighted Histogram Analysis Method (WHAM) (Grossfield's WHAM)[104,105], employing 50 bins along the reaction coordinate and a convergence tolerance of 0.0001 kcal/mol. Sufficient overlap between adjacent windows was verified to ensure reliable reconstruction of the free energy profile.

## Statistics and reproducibility

Statistical significance was assessed based on data parameters presented as the mean ± s.e.m. of $n$ independent cells or patches. The two-sided Student's $t$ test was used for comparing two groups, while a one-way ANOVA followed by Tukey's post hoc test was employed for pairwise comparisons involving more than two data groups. Statistical significance was highlighted as *$p < 0.05$, **$p < 0.01$.

### Reporting summary

Further information on research design is available in the Nature Portfolio Reporting Summary linked to this article.

## Data availability

Source data are available as a Source Data File, on figshare.com (https://doi.org/10.6084/m9.figshare.28304378), and additional data are provided in the Supplementary Information. The MD simulated structures in protein data bank (PDB) and protein structure file (PSF) formats as well as force field parameters, configuration files are available on figshare.com. The cryo-EM data used in this study are available at: https://doi.org/10.2210/pdb5VA1/pdb, https://doi.org/10.2210/pdb8EOW/pdb, https://doi.org/10.2210/pdb8EP0/pdb, and https://doi.org/10.2210/pdb8EP1/pdb for the PDB accession codes 5VA1, 8EOW, 8EP0, 8EP1. Source data are provided with this paper.

## Code availability

Standard codes for the main computational simulations can be accessed at: tcbg.illinois.edu. Scripts for the computational analysis of simulated data are available on figshare.com (https://doi.org/10.6084/m9.figshare.28304378).

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

## Acknowledgements

We thank Natalie L. Macchi for technical support; Joel C. Eissenberg for editing the manuscript; Enrico Di Cera and Nicola Pozzi for feedback, E. James Petersson and Venkatesh Yarra (University of Pennsylvania) for sharing the noncanonical amino acid Acd and providing advice; Ryan A. Mehl (GCE4All Research Center, Oregon State University) for sharing the Acd RS1 aminoacyl tRNA synthetase and tRNA pair plasmid. This research is supported by National Institutes of Health (NIH) grants R35GM154778 (to G.D.) and R56HL169176 (to G.D. and E.T.). We gratefully acknowledge support from the Doisy Fund of the Edward A. Doisy Department of Biochemistry and Molecular Biology at Saint Louis University School of Medicine. In addition, the authors acknowledge funding from the NIH grant R24GM145965 (to E.T.) and National Science Foundation (NSF) QCB grant: 221835 (to E.T.). Computing resources were provided by the NSF ACCESS award: MCA06N060 (to E.T.).

## Author contributions

G.D. conceived and designed the research. G.D. performed and analyzed fluorescence imaging experiments. L.J.H. performed patch-clamp recordings. L.J.H. and G.D. analyzed patch-clamp data. E.N.L. and G.D. performed molecular cloning of mutant hERG constructs. A.N.C, C.D.Q., E.T., and G.D. designed MD simulations. C.D.Q. performed and analyzed umbrella sampling of SMD simulations. A.N.C. performed and analyzed other MD simulations. E.T. supervised the MD simulations and computational analysis. G.D. wrote the manuscript with inputs from coauthors.

## Competing interests

The authors declare no competing interests.
