## [Transparent Peer Review file · Nature Communications]

Voltage Sensor Conformations Induced by LQTS-associated Mutations in hERG Potassium Channels

Corresponding Author: Dr Gucan Dai

Version 0:

Reviewer comments:

Reviewer #1

(Remarks to the Author)

The manuscript entitled 'Voltage Sensor Conformations Induced by LQTS-associated Mutations in hERG Potassium Channels' by Chan et al. studied the conformational dynamics of the hERG voltage sensor and structural effects of LQTS-associated mutations using cutting-edge techniques, including noncanonical amino acid incorporation, FLIM-transition metal FRET and MD simulations. The FLIM-tmFRET provides accurate FRET efficiency measurements reporting conformational changes and also allows FRET heterogeneity analyses reflecting the conformational landscapes, therefore offering a powerful approach to studying the conformational dynamics of hERG channel voltage sensors in native cell membranes. Moreover, the authors also conducted MD simulations on hERG voltage sensors carrying LQTS-associated mutations and produced very consistent results.

The present work identified an intermediate FRET conformation of hERG channel voltage sensors caused by LQTS mutation R531Q that may underlie its changes in channel function. However, identification of the intermediate conformational state in hERG channel voltage sensors, mainly by linear unmixing analyses of FLIM phasor plots, is quite weak. Similar analyses on the D540K mutant were conducted to support the conclusion of the intermediate S4 conformation, but the conclusion of the D540K voltage sensor exhibiting intermediate S4 conformation itself is also mainly based on FLIM phasor plot data.

For voltage-gated ion channels, numerous studies have already shown that there are multiple intermediate conformational states when voltage sensors transit from resting to activated states, so it is certainly not surprising to find that the gating charge mutations in hERG channels may stabilize one of these intermediate conformational states.

The major concern is that the present work also did not establish a clear structure-function relationship to explain how the intermediate conformational state in hERG voltage sensors was specifically induced by LQTS-associated mutations as the mechanism underlying their changes in the channel function.

Major comments:

1. The authors stated that R531 and R534 mutations are associated with LQTS, but their effects on G-V curves of hERG channels are very different. The R531Q mutation causes a right shift in the G-V curve, while the R534 mutant exhibits a left shift, and the R528Q has no shift. However, the results from the R534Q showed it did not induce similar intermediate S4 conformation, but the R528Q mutant did. These results do not support that the S4 intermediate conformation induced by LQTS-associated mutations is 'the structural underpinnings underlying voltage sensing in cardiac arrhythmias', as stated in the abstract. The MD simulation results also indicated that the R528Q and R531Q mutants exhibit similar intermediate S4 conformations, but no R543Q simulation results were included. I felt that the results of this study, with its current form, did not provide sufficient data for us to understand how LQTS mutations change the function of hERG channels to cause LQTS.

2. The D540K mutant was used as a proof-of-principle experiment to validate the efficacy of phasor analysis in uncovering conformational heterogeneity. However, except for the statement that 'D540K mutation introduces additional intermediate S4 conformations, likely contributing to the U-shaped behavior', no structural evidence was included to support the claim. Given that the R528Q mutation does not change the G-V curve but also induces intermediate S4 conformations (as shown by MD simulation), it is not appropriate to assume that the U-shaped G-V curve must be associated with intermediate S4 conformations. Therefore, using the D540K mutant as a proof-of-principle experiment is not logically sound or strong.

3. It is appreciated that this work also conducted FLIM-tmFRET studies on different labeling sites to validate their findings on the R531Q mutant. However, why FRET efficiencies from WT background at P440/L523 labeling sites, which are more distal than the G439/L523 labeling sites (Fig 4g), under all conditions, are much higher than those in Fig 1i, from G439/L523 labeling sites?

4. In fact, FLIM-tmFRET generates very accurate FRET efficiencies, which can predict the distances between FRET pair (Cu²⁺ and Acd in this manuscript). In Fig 4c, correlation analysis was performed at different labeling sites between the FRET-predicted distances and those from structures in the depolarizing condition. Since the up and down state structures of hERG channels are available, plus the intermediate state structures generated by MD simulation in the present work, more thorough correlation analyses should be included to compare the distances predicted by FLIM-tmFRET data with PBS (resting state) and 120 mM KCl (the depolarizing state) with those from structures at corresponding state. Similar correlation analyses can also be performed on the WT and R531Q mutation background to uncover whether the R531Q mutation alters the conformation landscapes of the hERG voltage sensor under different conditions.

5. Page 17, the authors stated that 'They suggest a novel mechanism through which LQTS-associated mutations can influence the gating of hERG channels and voltage sensing in cardiac tissue'. It is very unclear. Did the authors suggest that only LQTS-associated mutations give rise to intermediate S4 conformations, which is the novel mechanism? If so, how do the authors explain the results of the R534Q mutation, which does not induce intermediate S4 conformations? Perhaps it is reasonable to conclude that voltage-sensing arginine mutations may affect the voltage-dependent movement of the S4, but the results in this work do not support the hint that intermediate S4 conformations are linked with LQTS-associated mutations.

6. Page 17, '...can be attributed to the more left shift in the G-V relationship observed in R528Q channels (Supplementary Fig. 1)', should be '(Supplementary Fig. 1d).

Reviewer #2

(Remarks to the Author)

In this manuscript the authors investigate the gating dynamics of the hERG channels with respect to the central S4 helix. The down-movement of the S4 during hyperpolarization has been notoriously difficult to determine. They use a technical tour-de-force to determine the conformational changes upon depolarization (120 KCl) in hERG channels as well as when introducing the LQTS inducing mutant R531Q. To do this they perform FLIM-tmFRET on living cells transfected with a hERG channel construct containing two non-canonical amino acids (ncAAs). Both ncAAs are inserted at site-specific locations using a dual stop-codon suppression strategy. This is an experimental tour-de-force. The accomplishment is impressive.

Major issues:

1. Generally, I find the text very difficult to understand. The introduction is very limited with little info about structure-function relationships in hERG (or VGIC) channels. What is known about its overall structure? How does the S4 relate to the rest of the protein. What is known about the open- and closed conformations? This should be used to add significance to studying the up- and down conformations of the S4. The first paragraph of the Discussion could be beneficial in the Introduction.
2. Along those lines, the reason why this manuscript qualifies to be published here, is because of its FLIM-tmFRET application on live cells using ncAAs. A paragraph which underlines the significance and novelty of this would be beneficial.
3. The understanding of the results would also benefit greatly from adding a couple of explanatory sentences in the beginning of a section which explains the purpose of the performed experiments. What is the objective that these experiments will address? Then, after the described results, use a couple of sentences to provide interpretations about what the results would suggest about the biology of the system.
4. The authors state in the introduction that they wish to 'examine the impact of a LQTS-associated mutation (R531Q) on conformational rearrangements of hERG voltage sensors in transfected cell lines.' However, in figure 2 and 3 the authors investigate the D540K mutant as a proof-of-principle experiment for phasor analysis. It is not clear to me why a substantial amount of a central part of the paper is used in the investigation of the D540-R665 salt bridge. What is this a proof-of-principle of? Primarily, I do suggest that the authors are more elaborate on what the data should substantiate. If the data are required as proof-of-principle, then I suggest they go to SI so they do not intercept the flow in the main message.
5. It is not entirely clear to me what construct hERG WT+ is. In the legend for Fig. 1a WT+ is defined as hERG S620T. However, further in the legend for Fig. 1g, I understand WT+ is defined as hERG-WT with S620T and the L523TAG and G439TAA mutations. Then in Fig. 1h+I, I am in doubt if the R531Q mutation is made in the background of WT or WT+. I suggest that the authors find a consistent nomenclature for all background constructs and use that when introducing mutants. Instead of WT+ one could name hERG S620T-L523TAG-G439TAA as hERGTmFRET or WT+tmFRET.
6. The setup with insertion of two non-canonical amino acids and using it for FLIM-FRET is impressive. This combination of techniques is new to me and I do believe it would be inspiring to many. Its significance would require some fundamental controls, which I failed to find in the manuscript.
 - a. What is the background fluorescence in the absence of hERGTmFRET? This is assessed by co-transfection of PyIT/RS and the Acd RS1 tRNA synthetase plasmids along with Acd and ProK (+/-conjugation with Cu²⁺:DOTA).
 - b. What is the read-through for both the L523TAG and G439TAA mutations? This is determined by adding hERGTmFRET

expressing plasmid to a) and omitting either Acd or ProK, respectively. Then perform the current traces as in Fig 1a. The issue is that if ncAA insertion is within a sub-population of all expressed hERGs, then the measured G-V relationship from WT+ could be from read-through hERGs whereas the fluorescent signal originated from another hERG population.

7. I must admit that phasor plots are new to me. I found that the authors attempted to guide the reader in understanding the phasor plots with the following (page 7): "In the phasor plot of frequency domain FLIM, the lifetime data for each pixel in the confocal image were translated on a Cartesian coordinate, delineated by a universal semicircle (Fig. 1f)18,39,40. Fourier transformation was implemented; thus, the x-axis represents the real (G) component while the y-axis represents the imaginary (S) component of the lifetime decay information. Mono-exponential lifetimes precisely aligned with the semicircle, whereas multiexponential lifetimes localize within it. Shorter lifetimes demonstrate a clockwise shift along the semicircle." To understand this explanation one must know what a Cartesian coordinate is and the nature of the real (G) and imaginary (S) components. I could be the ignorant one here, but when submitting to a general journal as Nat Commun, one must expect that phasor plots will be new to a large proportion of the readers. Using Fig 1g-i as example, I understand (according to Wiki) that the blue line is a vector. It denotes what? Mean of the lifetime? What does the brown curved line provide of info? The data points are so small, that it is impossible to be subjective about them. The data in g) and i) look indistinguishable to me, yet the Apparent FRET is 0.15 and 0.10, respectively. Those differences are nicely depicted in i) – a plot I can easily relate to. I recommend that the authors plot their FLIM-tmFRET data differently. Showing them as changes in Acd lifetime as in f) would be illustrative if normalized. Alternatively, the phasor plots must be explained in much higher detail.

8. 'In addition, since the A430TAA site is more buried in the membrane, the labeling of Cu²⁺-DOTA might not be complete.' Please state how you assess complete labeling in any of the used sites.

Minor issues:

In Fig 2a: 'Distances between the β carbon of L523 and the α carbon of G439, D(C β -C α), are 31 Å and 22 Å in the down and up states, respectively.' What is this distance based upon? The MD simulations? Please state.

In Fig 2b: 'The low FRET state yields negligible FRET'. To assess this the QI of the inserted Acd donor alone must be determined. I cannot find that experiment.

In Fig 2b: 'The red cursor', Do you mean the black arrow?

Fig 3a-d: 'Representative analysis of the L523/G439 tmFRET from individual cells for hERG WT+, hERG-D540K, hERGD540K/R665E and hERG-R531Q channels under various conditions.' Any figure legend must be self-explanatory. Why are you showing this? What should the reader pay attention to?

Fig. 4c: As I understand the data, the A430TAA mutant goes against the linear trend observed by the others. This suggests that even though there in many cases is a linear relationship between distance and tmFRET efficiency, other factors, such as polarized fluorescence and dipole moments of acceptor do also contribute. Thus, using tmFRET as a molecular ruler must be done with caution.

'WT**' and 'WT*' in Suppl. Fig 2 and 4, respectively. Should they be renamed to WT+?

Reviewer #3

(Remarks to the Author)

In the present manuscript, Chan et al. used a combination of experimental and computational assays to characterize the effects of LQTS-associated mutations on the voltage sensor conformation of hERG channels. The scientific question is well posed and the experiments have been adequately conducted, although there are some gaps in the description of the methods and in the choice of systems studied.

The main issue with the manuscript is its novelty. Several results reported do not present sufficient novelty as they have already been published in other works that are reported below. The major contribution of this work to the scientific community seems primarily methodological, particularly regarding the FLIM/FRET sections, which are highly technical and difficult to understand for a general public.

For these reasons, I cannot recommend publication in Nature Communications in the current version. The main points to be addressed are as follows:

Electrophysiological sections:

1. The reason behind the choice of the mutants should be clarified. For example, in the first paragraph of the results, the effects of R531Q and other mutants such as R534L and R534Q are reported but it is not clear why for R531 the mutation was only to Q while for R534 to L and Q.
2. Did the authors study R531Q in the S620T mutant? Please clarify.
3. The results should be discussed in view of <https://doi.org/10.1007/s00232-013-9539-6> where the effects of R531Q and R531W mutations on the gating of the channel have already shown.

FLIM/FRET sections:

1. Why was the Acd site placed at L523 and the ProK/Cu²⁺-DOTA site at G439? This should be clarified because the choice is far from trivial. For example, why the reference point for S4 movements was not chosen to be F463, i.e., the usual gating charge transfer center?

2. The results of D540K and R665E mutants (page 11) are not novel; the authors should emphasize the difference between the present work and <https://doi.org/10.1074/jbc.M200410200>

MD simulations sections:

1. The computational models of hERG with the VSD in the closed configurations were already proposed in <https://doi.org/10.1038/s42003-022-03074-9>; the results and the structures should be carefully compared with this reference.
2. Previous simulation reports of the VSD movement of VGIC used free-energy or rare-event methods to overcome free energy barriers in the movement of S4 (see, e.g., <https://doi.org/10.1073/pnas.1416959112>, <https://doi.org/10.1085/jgp.201210827>, <https://www.nature.com/articles/s41467-024-45514-6>). It is rather surprising that the present simulations report S4 equilibration in unbiased simulations and at low voltages. Could the author expand on this point?
3. The major issue with the simulations seems to be the system that has been used - a monomer embedded in a lipid environment if I understand correctly. The biological and structural significance of this system is unclear as the S4 helix in this case is poised to move in a different environment from the physiological one.
4. The reason why the authors run additional SMD simulations is not clear. Did they observe the displacement of the VSD caused by the mutations in the equilibrium simulations (see also point 1)? If yes, what is the additional information provided by SMD?
5. The Methods lack adequate detail: did the authors simulate R531Q and R528Q mutants? How was computational mutagenesis performed? How did they check the protonation state of titratable residues? How did they embed the channel in the membrane? How many lipids of each type are there in the membrane? Which force field did they use? What was the equilibration protocol? How many replicas were simulated? Did all systems reach convergence after 1 μ s simulations?
6. The structural change in the alpha-helix should be characterized quantitatively.

Reviewer #4

(Remarks to the Author)

Version 1:

Reviewer comments:

Reviewer #1

(Remarks to the Author)

I appreciate that the authors have provided additional data and made substantial revisions to address the concerns raised in my initial review. I have no further comments on this manuscript.

Reviewer #2

(Remarks to the Author)

The additions and corrections have sufficiently strengthened the manuscript. I have no additional comments.

Reviewer #3

(Remarks to the Author)

The authors have added new data to their work, especially new simulations. Many of my previous comments have been resolved, but there are some points that still need to be clarified. However, I maintain my previous general opinion on the work: I do not question the quality of the work, but I believe it is not suitable for a general audience such as that of Nat Comm; the work is very technical and the novelty is purely methodological.

Major points:

- 1- A comparison of the closed structures from <https://doi.org/10.1038/s42003-022-03074-9> and the present ones was suggested. Rather, a comparison of the computational protocol was provided. In the cited reference, structures were produced according to experiments performed directly on hERG channels (<https://doi.org/10.1085/jgp.200409119>), while in the present work the authors ran the SMD simulations targeting the closed configuration of the VSD found by MacKinnon in EAG channels in 2022. For this reason it would be interesting to know whether the conceptually different protocols yield the same final structure. Also it would be necessary to report explicitly the gating charge of closed structure.
- 2- "His had single protonation in their side chains" is not an accurate statement that allows for reproducing the results. Histidines can have a single protonation but on different sites. Moreover, to allow the reproducibility of the structures, authors should clarify the software used to predict such protonation states.
- 3- In the methods, the authors stated "After equilibration, the WT tetramer system was simulated under an applied potential of -150 mV for 1 μ s," but the RMSD profiles shown in Fig. S11 show a simulation time of 250 ns. If the authors simulated several replicas for 250 ns, they should report the RMSD profile of each replica; if they simulated one replica for 1 μ s, they should report the entire simulation length. This mismatch should be appropriately clarified. We remind that the publisher has a specific policy on the number of replicas.

4- The inclusion of the tilt and distances of upward displacement of the S4 improved the manuscript; however, the authors did not provide details on their definition and on the calculation of such quantities.

5- I appreciate the new US simulations, but how did the authors assess the convergence of the PMFs? One could argue that 10 ns per window are not sufficient to obtain reliable results. On the other hand, in the past some discussion of the reaction coordinate was made (compare, eg, <https://doi.org/10.1073/pnas.1416959112>). In general, such free energy could be very informative, but we appreciate that this could be an endeavor beyond the scope of the manuscript.

Minor points:

1- The point that R531Q and R534L are associated with LQTS is not appropriately discussed in the text; the authors should clarify this in the introduction and then motivate the reason behind the choice of such mutations in the results.

2- "Point mutations were generated using the VMD Mutator Plugin" - reference is missing.

3- It is very difficult to follow the authors in the description of how many systems they simulated - with which thermodynamic conditions, how many replicas, the simulation time, the systems, etc. I strongly suggest the authors add a table in the SI summarizing all the simulations run.

Reviewer #4

(Remarks to the Author)

Reviewer #1 (Remarks to the Author):

The manuscript entitled 'Voltage Sensor Conformations Induced by LQTS-associated Mutations in hERG Potassium Channels' by Chan et al. studied the conformational dynamics of the hERG voltage sensor and structural effects of LQTS-associated mutations using cutting-edge techniques, including noncanonical amino acid incorporation, FLIM-transition metal FRET and MD simulations. The FLIM-tmFRET provides accurate FRET efficiency measurements reporting conformational changes and also allows FRET heterogeneity analyses reflecting the conformational landscapes, therefore offering a powerful approach to studying the conformational dynamics of hERG channel voltage sensors in native cell membranes. Moreover, the authors also conducted MD simulations on hERG voltage sensors carrying LQTS-associated mutations and produced very consistent results.

The present work identified an intermediate FRET conformation of hERG channel voltage sensors caused by LQTS mutation R531Q that may underlie its changes in channel function. However, identification of the intermediate conformational state in hERG channel voltage sensors, mainly by linear unmixing analyses of FLIM phasor plots, is quite weak. Similar analyses on the D540K mutant were conducted to support the conclusion of the intermediate S4 conformation, but the conclusion of the D540K voltage sensor exhibiting intermediate S4 conformation itself is also mainly based on FLIM phasor plot data.

For voltage-gated ion channels, numerous studies have already shown that there are multiple intermediate conformational states when voltage sensors transit from resting to activated states, so it is certainly not surprising to find that the gating charge mutations in hERG channels may stabilize one of these intermediate conformational states.

The major concern is that the present work also did not establish a clear structure-function relationship to explain how the intermediate conformational state in hERG voltage sensors was specifically induced by LQTS-associated mutations as the mechanism underlying their changes in the channel function.

We thank the reviewer for the positive comments on the approach of this new fluorescence/protein engineering strategy. The primary goal of this paper is to introduce a methodological advance—one that enables intramolecular distance measurements in living cells and native membranes. This approach, validated through the study of hERG voltage sensors in this work, offers broad applicability for investigating conformational rearrangements of transmembrane helices in membrane proteins.

Indeed, proposing intermediate states requires more evidence. To address this, we have performed additional molecular dynamics simulations, including steered MD of mutants and umbrella sampling, to characterize the conformational free energy landscape of S4 movement. In the revised manuscript (new Fig. 6), the resulting potential of mean force (PMF) profiles revealed distinct free energy minima for the wild-type and mutant voltage-sensing domains. These findings align with our FRET results, providing additional mechanistic support for the intermediate states and strengthening structure-function relationships.

Major comments:

1. The authors stated that R531 and R534 mutations are associated with LQTS, but their effects on G-V curves of hERG channels are very different. The R531Q mutation causes a right shift in the G-V curve, while the R534 mutant exhibits a left shift, and the R528Q has no shift. However, the results from the R534Q showed it did not induce similar intermediate S4 conformation, but the R528Q mutant did. These results do not support that the S4 intermediate conformation induced by LQTS-associated mutations is 'the structural underpinnings underlying voltage sensing in cardiac arrhythmias', as stated in the abstract. The MD simulation results also indicated that the R528Q and R531Q mutants exhibit similar intermediate S4 conformations, but no R543Q simulation results were included. I felt that the results of this study, with its current form, did not provide

sufficient data for us to understand how LQTS mutations change the function of hERG channels to cause LQTS.

We'd like to clarify that inducing an intermediate S4 conformation is only one mechanism, but not necessarily a universal mechanism underlying LQTS-associated mutations in S4. Our study focuses on specific arginine mutations within the gating charge transfer center of the voltage sensor, providing structural insights through a novel fluorescence-based approach combined with MD simulations, rather than suggesting a general mechanism for all LQTS-associated S4 mutations. The abstract has been revised accordingly.

R534 (R4) is not a primary contributor to the gating charge of hERG channels. A previous study (Zhang et al., JGP 2004; doi:10.1085/jgp.200409119, Fig. 4) demonstrated that only K525 (K1), R528 (R2), and R531 (R3) significantly contribute to the total gating charge, as neutralizing these residues reduced the overall gating charge per channel. In contrast, R534 is located in the lower portion of the S4 segment and may not reside or is only partially within the transmembrane electric field during S4 movement. It is not essential for the charge transfer center. Our revised mechanistic model also supports the idea that mutations within the charge transfer center, specifically R528 and R531, more strongly disrupt the salt bridge network within VSD, and produced tilting/bending of the S4 helix. Nevertheless, R534 may still play an important role in coupling the voltage sensor to pore gating.

To support our FRET findings, we also performed MD simulations of the R534Q VSD. In agreement with the experimental data, R534Q did not induce significant conformational changes in the S4 (Fig. S9), in contrast to the pronounced alterations observed with R531Q and R528Q. Overall, the absence of an intermediate state for R534Q remains consistent with our proposed mechanism.

2. The D540K mutant was used as a proof-of-principle experiment to validate the efficacy of phasor analysis in uncovering conformational heterogeneity. However, except for the statement that 'D540K mutation introduces additional intermediate S4 conformations, likely contributing to the U-shaped behavior', no structural evidence was included to support the claim. Given that the R528Q mutation does not change the G-V curve but also induces intermediate S4 conformations (as shown by MD simulation), it is not appropriate to assume that the U-shaped G-V curve must be associated with intermediate S4 conformations. Therefore, using the D540K mutant as a proof-of-principle experiment is not logically sound or strong.

The goal of using the D540K mutant and its charge swapping rescuing mutants D540K/R665E was to show that our FLIM-tmFRET approach was able to detect small distance changes. We removed the expression "proof-of-principle" to avoid confusion here, because it was only meant to showcase the technical aspect of our new methodology, not to suggest that the D540K experiment was for proof-of-principle demonstration of detecting intermediate states of S4. We did not aim to test there was an intermediate state for the D540K.

Our initial purpose of using the D540K mutant was to validate the power of detecting sub-nanometer distance changes using the FLIM-tmFRET/nonlinear unmixing strategy. The rationale here is that the U-shaped G-V suggests two distinct voltage-dependent gating processes, and different structural states of S4 at hyperpolarizing and depolarizing voltages. Not surprisingly, different FRET efficiencies were detected for D540K at hyperpolarizing and depolarizing voltages, in comparison to the single FRET efficiency for the WT (hERG_{tmFRET}). The D540K/R665E mutant behaves more like the WT (hERG_{tmFRET}), further validating the precision of this methodology.

The G-V relationship alone does not reliably indicate the presence of intermediate states in S4 as it is an ensemble averaging of voltage-dependent gating of many molecules and often shows no direct correlation.

However, the double-exponential activation kinetics observed for R528Q is consistent with the involvement of intermediate states in its gating mechanism.

We decided to move the D540K results to Supplementary Figures as also suggested by the Reviewer 2. Again, we want to clarify that the D540K experiments were designed, in addition to using different FRET pairs as we did in Fig 4, to validate the power of FLIM-tmFRET in detecting small distance changes, a very challenging task, and we are the first to be able to do that for membrane proteins in live-cell experiments.

3. It is appreciated that this work also conducted FLIM-tmFRET studies on different labeling sites to validate their findings on the R531Q mutant. However, why FRET efficiencies from WT background at P440/L523 labeling sites, which are more distal than the G439/L523 labeling sites (Fig 4g), under all conditions, are much higher than those in Fig 1i, from G439/L523 labeling sites?

We believe this confusion arises from the two different strategies for FRET estimation in the phasor plot: FRET trajectory fitting (only in Fig 1) and linear unmixing fitting (other Figures). The trajectory fitting method leads to an underestimation of FRET, which we have further clarified in the paper (also through a new supplementary Fig S3). This limitation motivated us to use the linear unmixing method instead, which generates more accurate distance estimation. Comparisons of different FRET pairs should be done within the same method. As shown in Fig. 4c, the FRET efficiency measured using the linear unmixing method for the P440/L523 pair is lower than that of the G439/L523 pair, consistent with the longer distance between P440 and L523 sites.

4. In fact, FLIM-tmFRET generates very accurate FRET efficiencies, which can predict the distances between FRET pair (Cu²⁺ and Acd in this manuscript). In Fig 4c, correlation analysis was performed at different labeling sites between the FRET-predicted distances and those from structures in the depolarizing condition. Since the up and down state structures of hERG channels are available, plus the intermediate state structures generated by MD simulation in the present work, more thorough correlation analyses should be included to compare the distances predicted by FLIM-tmFRET data with PBS (resting state) and 120 mM KCl (the depolarizing state) with those from structures at corresponding state. Similar correlation analyses can also be performed on the WT and R531Q mutation background to uncover whether the R531Q mutation alters the conformation landscapes of the hERG voltage sensor under different conditions.

Agree. We have incorporated new analyses (Fig. 4b) regarding the intermediate I (PBS) and the up state (120 mM KCl) of the R531Q in the revised manuscript. We compared the FRET-predicted distances and compared those with MD simulation derived distances side by side, and they generated similar changes in distance.

5. Page 17, the authors stated that 'They suggest a novel mechanism through which LQTS-associated mutations can influence the gating of hERG channels and voltage sensing in cardiac tissue'. It is very unclear. Did the authors suggest that only LQTS-associated mutations give rise to intermediate S4 conformations, which is the novel mechanism? If so, how do the authors explain the results of the R534Q mutation, which does not induce intermediate S4 conformations? Perhaps it is reasonable to conclude that voltage-sensing arginine mutations may affect the voltage-dependent movement of the S4, but the results in this work do not support the hint that intermediate S4 conformations are linked with LQTS-associated mutations.

We appreciate the reviewer's insightful comment. To clarify, we did not intend to imply that all LQTS-associated mutations in S4 induce intermediate conformations. As the reviewer correctly notes, our data show that the R534Q mutation does not introduce additional S4 states. Instead, our findings suggest that mutations of key arginines involved in the charge transfer center stabilize intermediate conformations, indicating a distinct mechanism by which specific mutations disrupt voltage sensing. To prevent any confusion, we have revised the text to emphasize that intermediate S4 conformations are not a general feature of all LQTS-associated

mutations, but are specific to mutations of the primary arginines critical for voltage sensing.

6. Page 17, '...can be attributed to the more left shift in the G-V relationship observed in R528Q channels (Supplementary Fig. 1)', should be '(Supplementary Fig. 1d).

Corrected.

Reviewer #2 (Remarks to the Author):

In this manuscript the authors investigate the gating dynamics of the hERG channels with respect to the central S4 helix. The down-movement of the S4 during hyperpolarization has been notoriously difficult to determine. They use a technical tour-de-force to determine the conformational changes upon depolarization (120 KCl) in hERG channels as well as when introducing the LQTS inducing mutant R531Q. To do this they perform FLIM-tmFRET on living cells transfected with a hERG channel construct containing two non-canonical amino acids (ncAAs). Both ncAAs are inserted at site-specific locations using a dual stop-codon suppression strategy. This is an experimental tour-de-force. The accomplishment is impressive.

We thank the reviewer for the positive comments and helpful reviews.

Major issues:

1. Generally, I find the text very difficult to understand. The introduction is very limited with little info about structure-function relationships in hERG (or VGIC) channels. What is known about its overall structure? How does the S4 relate to the rest of the protein. What is known about the open- and closed conformations? This should be used to add significance to studying the up- and down conformations of the S4. The first paragraph of the Discussion could be beneficial in the Introduction.

The introduction has been expanded by adding a new section on the structure-function relationships in hERG.

2. Along those lines, the reason why this manuscript qualifies to be published here, is because of its FLIM-tmFRET application on live cells using ncAAs. A paragraph which underlines the significance and novelty of this would be beneficial.

New paragraph added to the introduction to highlight the significance and novelty of this new method.

3. The understanding of the results would also benefit greatly from adding a couple of explanatory sentences in the beginning of a section which explains the purpose of the performed experiments. What is the objective that these experiments will address? Then, after the described results, use a couple of sentences to provide interpretations about what the results would suggest about the biology of the system.

New explanatory sentences now added to the results.

4. The authors state in the introduction that they wish to 'examine the impact of a LQTS-associated mutation (R531Q) on conformational rearrangements of hERG voltage sensors in transfected cell lines.' However, in figure 2 and 3 the authors investigate the D540K mutant as a proof-of-principle experiment for phasor analysis. It is not clear to me why a substantial amount of a central part of the paper is used in the investigation of the D540-R665 salt bridge. What is this a proof-of-principle of? Primarily, I do suggest that the authors are more elaborate on what the data should substantiate. If the data are required as proof-of-principle, then I suggest they go to SI so they do not intercept the flow in the main message.

We agree with the reviewer and have moved this part to the Supplementary figures.

We clarify that the D540K and charge-swapping D540K/R665E mutants were used to demonstrate that FLIM-tmFRET can detect small distance changes, not as proof-of-principle for detecting intermediate S4 states. To avoid confusion, we removed the word “proof-of-principle,” as it referred specifically to the methodological validation. Our rationale for using D540K was to validate the ability of FLIM-tmFRET/nonlinear unmixing to resolve sub-nanometer distance changes. The U-shaped G-V of D540K suggests distinct voltage-dependent gating processes, reflected in the different FRET efficiencies at hyperpolarizing and depolarizing voltages. The D540K/R665E mutant behaved more like WT+, further supporting the method’s precision.

5. It is not entirely clear to me what construct hERG WT+ is. In the legend for Fig. 1a WT+ is defined as hERG S620T. However, further in the legend for Fig. 1g, I understand WT+ is defined as hERG-WT with S620T and the L523TAG and G439TAA mutations. Then in Fig. 1h+I, I am in doubt if the R531Q mutation is made in the background of WT or WT+. I suggest that the authors find a consistent nomenclature for all background constructs and use that when introducing mutants. Instead of WT+ one could name hERG S620T-L523TAG-G439TAA as hERGtmFRET or WT+tmFRET.

We have renamed these constructs based on the suggestions, called hERG_{tmFRET}.

6. The setup with insertion of two non-canonical amino acids and using it for FLIM-FRET is impressive. This combination of techniques is new to me and I do believe it would be inspiring to many. Its significance would require some fundamental controls, which I failed to find in the manuscript.

a. What is the background fluorescence in the absence of hERG_{tmFRET}? This is assessed by co-transfection of PyIT/RS and the Acd RS1 tRNA synthetase plasmids along with Acd and ProK (+/-conjugation with Cu²⁺:DOTA).

In the absence of co-expressed hERG_{tmFRET}, the background signal is generally weak compared to its presence, primarily due to the (1) ambient light background (infinite lifetime), (2) endogenous nonspecific emission (~420–480 nm) from the cytosol of tsA cells, and (3) nonspecific Acd fluorescence of the cytosol.

Since we strictly select only cells with bright, membrane-localized YFP signal for Acd imaging, specific fluorescence dominates, minimizing background fluorescence contamination. Additionally, background fluorescence is unlikely to affect our results because the donor Acd phasor distribution (without Cu²⁺:DOTA) is circular rather than oval, indicating little lifetime heterogeneity due to FRET or other types of quenching. Notably, the oval-shaped distribution is most pronounced under depolarized conditions when using the hERG_{tmFRET} construct.

We have used effective strategies to isolate membrane-localized Acd signal specific to the channel. As shown in Fig. S2b, weak background fluorescence (dim ambient light with an infinite lifetime) can be removed using intensity-based filtering. Additionally, membrane-localized Acd fluorescence was distinguished from cytosolic fluorescence by intensity-based filtering, guided by the known positions of membrane-localized YFP where hERG channels are expressed (Fig. S2d). In the absence of the acceptor Cu²⁺:DOTA, the lifetime of membrane-localized Acd remained largely unchanged under conditions such as high KCl treatment.

Our linear unmixing protocol estimated background contribution to be approximately 2–5% for both donor-alone and FRET samples, as reported in the source data file. While the background position in the phasor space could influence the estimated FRET efficiency (we used a (0,0) position for simplicity), the primary goal

of our FLIM-FRET analysis is to detect relative distance changes at high resolution rather than absolute distances. Refining this approach remains an area for future improvement.

b. What is the read-through for both the L523TAG and G439TAA mutations? This is determined by adding hERG_{tmFRET} expressing plasmid to a) and omitting either Acd or ProK, respectively. Then perform the current traces as in Fig 1a. The issue is that if ncAA insertion is within a sub-population of all expressed hERGs, then the measured G-V relationship from WT+ could be from read-through hERGs whereas the fluorescent signal originated from another hERG population.

We relied on visualizing fluorescence of the YFP fused to the C-terminal end of the hERG channel to indicate expression of full-length channel and to identify positive cells. Control experiments (Fig. S2) indicate minimal read-through at these sites. When ProK synthetase was omitted, membrane-localized YFP fluorescence was negligible, suggesting that the TAA site has little read-through or is accessible to the Acd system. Additionally, the nearly linear correlation between YFP fluorescence and membrane localized Acd levels suggests minimal read-through at the TAG site—if read-through were substantial, this correlation would not be observed. We previously conducted hERG_{tmFRET} experiments using synthetases but without Acd and ProK and observed no detectable currents—similar to untransfected HEK cells—even in the presence of TEA in the bath solution (this information has been added to the method section).

7. I must admit that phasor plots are new to me. I found that the authors attempted to guide the reader in understanding the phasor plots with the following (page 7): “In the phasor plot of frequency domain FLIM, the lifetime data for each pixel in the confocal image were translated on a Cartesian coordinate, delineated by a universal semicircle (Fig. 1f)^{18,39,40}. Fourier transformation was implemented; thus, the x-axis represents the real (G) component while the y-axis represents the imaginary (S) component of the lifetime decay information. Mono-exponential lifetimes precisely aligned with the semicircle, whereas multiexponential lifetimes localize within it. Shorter lifetimes demonstrate a clockwise shift along the semicircle.” To understand this explanation one must know what a Cartesian coordinate is and the nature of the real (G) and imaginary (S) components. I could be the ignorant one here, but when submitting to a general journal as Nat Commun, one must expect that phasor plots will be new to a large proportion of the readers. Using Fig 1g-i as example, I understand (according to Wiki) that the blue line is a vector. It denotes what? Mean of the lifetime? What does the brown curved line provide of info? The data points are so small, that it is impossible to be subjective about them. The data in g) and i) look indistinguishable to me, yet the Apparent FRET is 0.15 and 0.10, respectively. Those differences are nicely depicted in i) – a plot I can easily relate to. I recommend that the authors plot their FLIM-_{tmFRET} data differently. Showing them as changes in Acd lifetime as in f) would be illustrative if normalized. Alternatively, the phasor plots must be explained in much higher detail.

We have included expanded views of the phasor plot as insets. The 0.15 FRET one has the lifetime species more shifted (clockwise) relative to the donor alone (green cursor). Furthermore, we have provided a new supplementary figure (Fig. S3) that details the FRET trajectory method in the phasor plot and compares it with the linear unmixing method. In a FLIM phasor plot, the lines that connect different lifetime species to perform linear decomposition are “linear combination lines.” The position of a phasor point along the line reflects the fractional contribution of each species in the mixture.

8. ‘In addition, since the A430TAA site is more buried in the membrane, the labeling of Cu²⁺-DOTA might not be complete.’ Please state how you assess complete labeling in any of the used sites.

We tested different durations of click chemistry labeling (30 min, 1 hour, and 1.5 hours) to assess completeness and found that the Acd lifetime shift plateaued around 1.5 hours, suggesting near-complete labeling. Since we did not pursue A430TAA due to its arc-shaped phasor distribution, we did not investigate its

labeling time requirements. This sentence about A430 is rather speculative and has been moved to the Methods section instead.

In general, incomplete labeling of the FRET acceptor via azide-alkyne click chemistry affects the percentage of FRET but not FRET efficiency. Importantly, it does not alter the relative change in the percentage of the FRET state when imaging the same cell at different voltages. Therefore, we do not expect incomplete labeling of the FRET acceptor to impact the conclusions of this study.

Minor issues:

In Fig 2a: 'Distances between the β carbon of L523 and the α carbon of G439, D(C β -C α), are 31 Å and 22 Å in the down and up states, respectively.' What is this distance based upon? The MD simulations? Please state.

These are from our homology structural models based on the cryo-EM structures of hERG channels in liposome (Mandala and Mackinnon, PNAS, 2022). AlphaFold was also used to help build the S1-S2 loop region, which was unsolved in the cryo-EM structure of hERG (Wang and Mackinnon, Cell, 2017). We have revised this information in the paper.

In Fig 2b: 'The low FRET state yields negligible FRET'. To assess this the QI of the inserted Acd donor alone must be determined. I cannot find that experiment.

This is shown as the lifetime species nearly overlap with green colored cursor (donor Acd alone), suggesting little change in lifetime at 107 mM NMDG conditions. Throughout the paper, the green cursor in the phasor plot always indicates the donor alone in the absence of acceptors. FRET efficiency was measured as shift of the donor lifetime compared to donor alone. The phasor plot of donor alone is in Fig. S2.

In Fig 2b: 'The red cursor', Do you mean the black arrow?

Yes, and corrected: "red cursor on the semicircle". The larger red cursor is the lifetime position with background contribution.

Fig 3a-d: 'Representative analysis of the L523/G439 tmFRET from individual cells for hERG WT+, hERG-D540K, hERGD540K/R665E and hERG-R531Q channels under various conditions.' Any figure legend must be self-explanatory. Why are you showing this? What should the reader pay attention to?

Refined this part.

Fig. 4c: As I understand the data, the A430TAA mutant goes against the linear trend observed by the others. This suggests that even though there in many cases is a linear relationship between distance and tmFRET efficiency, other factors, such as polarized fluorescence and dipole moments of acceptor do also contribute. Thus, using tmFRET as a molecular ruler must be done with caution.

The arc shaped lifetime is due to the proximity of the FRET pairs even in the S4 down states, generating two FRET states representing S4 up and down states. It still follows the linear law of phasor addition but unmixing them is more difficult to do. In contrast, the hERG-WT_{tmFRET} (G439TAA) design has the FRET pairs far away when S4 is in the down state, allowing just showing one FRET state representing the S4 up state to be dominant. We were still able to perform two linear unmixing in one phasor plot for the A430TAA mutant in Fig S3, as compared to the single one for the hERG-WT_{tmFRET}.

'WT**' and 'WT*' in Suppl. Fig 2 and 4, respectively. Should they be renamed to WT+?

Yes, corrected.

Reviewer #3 (Remarks to the Author):

In the present manuscript, Chan et al. used a combination of experimental and computational assays to characterize the effects of LQTS-associated mutations on the voltage sensor conformation of hERG channels. The scientific question is well posed and the experiments have been adequately conducted, although there are some gaps in the description of the methods and in the choice of systems studied.

The main issue with the manuscript is its novelty. Several results reported do not present sufficient novelty as they have already been published in other works that are reported below. The major contribution of this work to the scientific community seems primarily methodological, particularly regarding the FLIM/FRET sections, which are highly technical and difficult to understand for a general public.

For these reasons, I cannot recommend publication in Nature Communications in the current version. The main points to be addressed are as follows:

We appreciate the reviewer's feedback and reminding us of previous citations and prior work to discuss. The FLIM-FRET method sections have also been further clarified and explained, making it easier for general audience. In addition, we have strengthened the manuscript—particularly the structure-function correlation of voltage sensing in hERG channels.

While our paper is built on previous findings (particularly LQTS mutations), the key innovation of our study is in its methodological advancements (FLIM-FRET in combination with MD simulations), which provide new structural insights beyond traditional electrophysiological approaches. Our work introduces a novel fluorescence-based methodology with broad application for studying membrane protein conformations:

- (1) This is the first study to use a noncanonical fluorescent amino acid (Acid), dual stop-codon suppression for site-specific fluorescence labeling, combined with phasor FLIM-tmFRET to measure intra-molecular distances in living cells with angstrom precision.
- (2) We pioneered the application of phasor FLIM-based linear unmixing in membrane protein structural biology, allowing simultaneous estimation of FRET efficiency and the fraction of molecules in the FRET state—capabilities previously unattainable in live-cell ensemble FRET studies.
- (3) The additional information provided by the fraction of FRET state from the phasor FLIM analysis allowed us to estimate the FRET-derived macroscopic free energy of activation for voltage-gated channels using Boltzmann's law—a conceptual and technical advance achieved in live-cell ensemble FRET experiments.

Given the lack of tools for probing conformational heterogeneity in live-cell membranes, our combined approach establishes an efficient platform for studying dynamic structural rearrangements in membrane proteins. These methodological advancements make our study impactful to the fields of fluorescence imaging and structural biology.

Electrophysiological sections:

1. The reason behind the choice of the mutants should be clarified. For example, in the first paragraph of the results, the effects of R531Q and other mutants such as R534L and R534Q are reported but it is not clear why for R531 the mutation was only to Q while for R534 to L and Q.

We have clarified this point in the revised text. R531Q and R534L are both known LQTS-associated mutations. R531L has not been found to be a LQTS associated mutation. R531Q allows us to assess the functional consequences of neutralizing the positive charge at this position with minimal structural disruption. The R-to-Q

mutation is also commonly observed in disease-associated mutations of voltage sensors in other voltage-gated channels, which is why we also tested the R534Q mutation.

2. Did the authors study R531Q in the S620T mutant? Please clarify.

Yes, we have clarified that all hERG constructs in this paper has the S620T mutation and name the R531Q as R531Q+.

3. The results should be discussed in view of <https://doi.org/10.1007/s00232-013-9539-6> where the effects of R531Q and R531W mutations on the gating of the channel have already shown.

We were aware of this paper and have cited it. This paper is purely electrophysiological and doesn't include any detailed structural mechanisms. The goal of our paper is not to repeat those findings but rather developing a novel method to reveal the underlying structural mechanisms.

FLIM/FRET sections:

1. Why was the Acd site placed at L523 and the ProK/Cu²⁺-DOTA site at G439? This should be clarified because the choice is far from trivial. For example, why the reference point for S4 movements was not chosen to be F463, i.e., the usual gating charge transfer center?

F463 was not chosen as the reference site because it is deeply embedded in the membrane and inaccessible to extracellularly applied azido-DOTA. Instead, we selected a site more exposed to the extracellular environment. Additionally, L523 was selected as the donor site because it is functionally non-critical and located far from the acceptor in the resting state, resulting in negligible FRET. Upon activation, it moves within the tmFRET detection range, enabling sensitive measurement of conformational changes. We also tested L520 as a donor site, but it did not produce robust channel expression, making L523 the optimal choice.

2. The results of D540K and R665E mutants (page 11) are not novel; the authors should emphasize the difference between the present work and <https://doi.org/10.1074/jbc.M200410200>

We have cited this paper, which first identified these mutations and the U-shaped G-V of D540K. In our study, D540K and R665E were used as a tool to validate the sensitivity of our FLIM-tmFRET approach in detecting small distance changes. To maintain the focus on the main findings, we have moved the D540K/R665E data to the supplementary section, as these mutants primarily serve as methodological controls.

MD simulations sections:

1. The computational models of hERG with the VSD in the closed configurations were already proposed in <https://doi.org/10.1038/s42003-022-03074-9>; the results and the structures should be carefully compared with this reference.

We have added a section to Discussion to compare our findings with those of the previous study. Our supporting MD simulations offer several important advances over that previous work:

(1) The previous study was conducted before the publication of the S4 down-state (hyperpolarized voltage, resting state) cryo-EM structure of Eag channels from the Mackinnon lab in October 2022. Previous cryo-EM structures of this family of channels were all in the S4 up-state solved in the absence of voltage (0 mV). In contrast, our equilibrium MD and steered MD (SMD) simulations of the hERG S4 down conformation are grounded in homology models derived from these recent structural data. This allowed us to model the S4 down conformation with significantly improved accuracy.

(2) The earlier study primarily focused on the coupling between the S4 voltage sensor and the pore domain, rather than on intermediate conformations of S4 or its dynamic transitions. Their SMD simulations were very short (2–4 ns) and the down-state S4 conformation was obtained by employing a simplistic pulling protocol targeting the center of mass of the S4 helix. As a result, their approach lacked the temporal and structural resolutions needed to capture the conformational transitions of S4. In contrast, our simulations were conducted over 100 ns for SMD and >250 ns for equilibrium MD simulations and used collective variables derived from the solved down-state cryo-EM structures of closely related Eag channels, enabling a mechanistically informative analysis. We also performed umbrella sampling to reveal free energy landscapes of the S4 movement in both WT and the mutants.

(3) The previous study did not present a clear picture of any significant structural distinctions or intermediate conformations in S4 between the wild-type and LQTS mutants. This limitation likely stemmed from the lack of structural guidance (i.e., absence of Eag-based homology models in their resting states) and the use of rapid, coarse SMD protocols. In our work, the use of structurally informed homology models and extended, higher-resolution simulations allowed us to resolve clear conformational differences between WT and mutant channels, providing new insight into the molecular basis of hERG channel dysfunction in LQTS.

In summary, our MD simulations, which align well with experimental FRET data, provide important new insights into the voltage-sensing mechanisms of hERG channels.

2. Previous simulation reports of the VSD movement of VGIC used free-energy or rare-event methods to overcome free energy barriers in the movement of S4 (see, e.g., <https://doi.org/10.1073/pnas.1416959112>, <https://doi.org/10.1085/jgp.201210827>, <https://www.nature.com/articles/s41467-024-45514-6>). It is rather surprising that the present simulations report S4 equilibration in unbiased simulations and at low voltages. Could the author expand on this point?

We have expanded on this point. Since we saw similar FRET efficiencies at high voltages but different FRET efficiency at low voltages due to S4 mutations, we were more interested in performing an unbiased simulation at low voltages to provide structural insights. This is also guided by homology modeling of the new cryo-EM structures of the related Eag channels solved at hyperpolarized voltages (2022), which was unavailable in previously published simulations.

3. The major issue with the simulations seems to be the system that has been used - a monomer embedded in a lipid environment if I understand correctly. The biological and structural significance of this system is unclear as the S4 helix in this case is poised to move in a different environment from the physiological one.

In the initial submission, we only presented the tetramer for the WT in a lipid environment. In the resubmission, we have added new simulations for the mutant channels in a tetrameric configuration in the same lipid environment and compared those with models in the monomer settings. They generated consistent results.

We found the R531Q altered the structural of the S4 helix similarly for the tetramer hERG channel versus the isolated monomer transmembrane domains, specifically exhibiting an upward shift and a tilt of the S4 helix. We have summarized these findings in the Fig 4a. Thus, the following SMD were using the monomer setting.

In general, the voltage-sensing domain (VSD) of ion channels can function as a modular, independent unit. This is exemplified in hERG and Eag channels, where split constructs with covalently disconnected VSDs and pore domains can reassemble into functional channels (Lörinczi et al., 2015, DOI: 10.1038/ncomms7672). For

Eag channels, even when both the entire N- and C-terminal regions are deleted, the channel still retain most depolarization-dependent activation properties (Whicher and Mackinnon, 2019, DOI: 10.7554/eLife.49188). Moreover, VSDs can be repurposed in other biological contexts, such as voltage-sensitive phosphatases (VSP), where they are fused to catalytic domains. Similarly, Hv proton channels are essentially homologous to standalone VSD monomers. These examples underscore the functional autonomy of the VSD, supporting the relevance of insights gained from monomeric VSD MD simulations.

4. The reason why the authors run additional SMD simulations is not clear. Did they observe the displacement of the VSD caused by the mutations in the equilibrium simulations (see also point 1)? If yes, what is the additional information provided by SMD?

Yes, there was a displacement and a considerable tilt of the S4 caused by R531Q. The purpose of the SMD simulations was to explore potential intermediate states of the VSD and to assess how specific mutations alter these conformations. To strengthen these findings, we conducted additional SMD simulations on the mutants and umbrella sampling simulations. They were used to generate conformational free energy landscape for both wild-type and mutant VSDs. These profiles not only support the conclusions drawn from our FRET experiments but also provide energetic insights into S4 movement and how it is disrupted by the mutations.

5. The Methods lack adequate detail: did the authors simulate R531Q and R528Q mutants? How was computational mutagenesis performed? How did they check the protonation state of titratable residues? How did they embed the channel in the membrane? How many lipids of each type are there in the membrane? Which force field did they use? What was the equilibration protocol? How many replicas were simulated? Did all systems reach convergence after 1 μ s simulations?

We now have added these details and improved the method section.

6. The structural change in the alpha-helix should be characterized quantitatively.

We have added quantitative descriptions on the structural change in the alpha-helix, comparing the WT and the mutants, particularly the degree of tilt and distances of upward displacement of the S4.

Reviewer #4 (Remarks to the Author):

Reviewer #1 (Remarks to the Author):

I appreciate that the authors have provided additional data and made substantial revisions to address the concerns raised in my initial review. I have no further comments on this manuscript.

Reviewer #2 (Remarks to the Author):

The additions and corrections have sufficiently strengthened the manuscript. I have no additional comments.

Reviewer #3 (Remarks to the Author):

The authors have added new data to their work, especially new simulations. Many of my previous comments have been resolved, but there are some points that still need to be clarified. However, I maintain my previous general opinion on the work: I do not question the quality of the work, but I believe it is not suitable for a general audience such as that of Nat Comm; the work is very technical and the novelty is purely methodological.

Major points:

1- A comparison of the closed structures from <https://doi.org/10.1038/s42003-022-03074-9> and the present ones was suggested. Rather, a comparison of the computational protocol was provided. In the cited reference, structures were produced according to experiments performed directly on hERG channels (<https://doi.org/10.1085/jgp.200409119>), while in the present work the authors ran the SMD simulations targeting the closed configuration of the VSD found by MacKinnon in EAG channels in 2022. For this reason it would be interesting to know whether the conceptually different protocols yield the same final structure. Also it would be necessary to report explicitly the gating charge of closed structure.

We thank the reviewer's thoughtful comments. The comparison between our MD simulations and those from the previous study, both in terms of technical aspects and underlying biological mechanisms, has been addressed in our previous response. Our simulation protocol represents a significant improvement. Moreover, the integration of experimental FRET imaging with MD simulations establishes this work as a benchmark for investigating membrane protein rearrangements, particularly ion channel macromolecules, in live cells. Although our study has limitations due to that a down-state cryo-EM structure of the hERG VSD is not yet available, it provides, for the first time, a detailed view of hERG channel voltage sensing and a novel structural mechanism for disease. In addition, our homology modeling and MD simulations on hERG VSD based on EAG channels are well aligned with FERT results. Further exploration of how the voltage sensor is coupled to the channel pore opening and closing remains an important area for future research beyond current scope.

2- "His had single protonation in their side chains" is not an accurate statement that allows for reproducing the results. Histidines can have a single protonation but on different sites. Moreover, to allow the reproducibility of the structures, authors should clarify the software used to predict such protonation states.

In the revised manuscript, we now specify the protonation states of histidine residues, the protonation only occurred on the δ -nitrogen (HSD). To ensure clarity and reproducibility, we have also clarified the software "VMD plugin psfgen" used to assign protonation states. These details have been added to the Methods section.

3- In the methods, the authors stated "After equilibration, the WT tetramer system was simulated under an applied potential of -150 mV for 1 μ s," but the RMSD profiles shown in Fig. S11 show a simulation time of 250 ns. If the authors simulated several replicas for 250 ns, they should report the RMSD profile of each replica; if they simulated one replica for 1 μ s, they should report the entire simulation length. This mismatch should be appropriately clarified. We remind that the publisher has a specific policy on the number of replicas.

We have provided further explanation and clarified this point in the Methods. In all equilibrium simulations, 250 ns is sufficient to generate convergence. The WT tetramer was initially simulated for 1 μ s, but found 250 ns was efficiently generate convergence, no significant changes in RMSD from 250 ns to 1 μ s (see the figure below).

Thus, all equilibrium MD simulations were run for 250 ns, except for an initial 1 μ s test run of the wild-type tetramer. For consistency of figure presentation, we only used 250 ns in Fig S11. The RMSD profiles were consistent across all independent replicas, indicating clear convergence. Since the goal is to demonstrate structural stability, the agreement between at least two independent replicas provides strong evidence.

4- The inclusion of the tilt and distances of upward displacement of the S4 improved the manuscript; however, the authors did not provide details on their definition and on the calculation of such quantities.

We have added in the figure legend that the helix angles were estimated using the PyMOL plugin `AngleBetweenHelices`: <https://pymolwiki.org/index.php/AngleBetweenHelices>

5- I appreciate the new US simulations, but how did the authors assess the convergence of the PMFs? One could argue that 10 ns per window are not sufficient to obtain reliable results. On the other hand, in the past some discussion of the reaction coordinate was made (compare, eg, <https://doi.org/10.1073/pnas.1416959112>). In general, such free energy could be very informative, but we appreciate that this could be an endeavor beyond the scope of the manuscript.

To assess convergence in PMFs, we monitored the stability of the PMFs over time by comparing results obtained from progressively shorter sampling segments toward the end of the simulation in each window, and we observed minimal variation, suggesting reasonable convergence within the sampling time. While we acknowledge that 10 ns per window may be on the shorter side, our results showed consistency across adjacent windows and satisfactory overlap between histograms, indicating that the sampling was sufficient for the purposes of this study. Regarding the choice of reaction coordinate, we are aware of the importance of selecting an appropriate collective variable, as highlighted in the literature. In our case, we chose a coordinate that captures the primary up-down movement of S4 helix in the voltage sensor activation. We agree that a more exhaustive exploration of alternative coordinates, e.g. rotation of helix, and extended sampling could yield additional insight. Such an endeavor would go beyond the scope of the current paper.

Minor points:

1- The point that R531Q and R534L are associated with LQTS is not appropriately discussed in the text; the authors should clarify this in the introduction and then motivate the reason behind the choice of such mutations in the results.

We have clarified this further in the text regarding R534. Only three arginine residues—R528, R531, and R534—are located within or near the charge transfer center, making all three worth studying.

2- "Point mutations were generated using the VMD Mutator Plugin" - reference is missing.

Reference added.

3- It is very difficult to follow the authors in the description of how many systems they simulated - with which

thermodynamic conditions, how many replicas, the simulation time, the systems, etc. I strongly suggest the authors add a table in the SI summarizing all the simulations run.

We have added further clarification in the Methods section regarding the simulation setup. Specifically, we have addressed the distinction between the 1 μ s and 250 ns durations for the tetramer MD simulations. While we considered including a summary table in the Supplementary Information, we concluded that the level of detail did not warrant a standalone table. However, we believe the revised text now clearly outlines the simulation parameters.

Reviewer #4 (Remarks to the Author):
